# Reflecting and Linking knowledge: Dynamic Label Structures for Prompt-based Continual Learning

## Abstract

Humans experience the world as a series of connected events, which can be organized hierarchically based on their conceptual knowledge. Drawing from this cognitive insight, we explore how our natural ability to organize and relate information can revolutionize the training of deep learning models. Our novel approach directly addresses the challenge of catastrophic forgetting by *leveraging the relationships within continuously emerging class data*. In particular, by creating a tree structure from an expanding set of labels, we uncover fresh perspectives on the data relationship, pinpointing groups of similar classes that easily lead to confusion. Additionally, we dive deeper into the hidden connections between classes by analyzing the behavior of the original pretrained model via an optimal transport-based approach. From these revelations, we propose a novel regularization loss function that encourages models to focus on challenging areas of knowledge, effectively boosting performance. Our experimental results demonstrate our effectiveness across a range of Continual learning benchmarks, paving the way for more effective AI systems. Our code is available at
`https://anonymous.4open.science/r/RefCL-EFC5/`.

## 1 Introduction

Continual Learning (CL) (Wang et al., 2024; Lopez-Paz & Ranzato, 2017) is a research direction that focuses on realizing the human dream of creating truly intelligent systems, where machines can learn on the go, accumulate knowledge, and operate in constantly changing environments as a human's companion. Despite the impressive capabilities of A.I systems, Continual Learning remains a challenging scenario due to the tendency to forget obtained knowledge when facing new ones, known as *catastrophic forgetting* (French, 1999). In dealing with this challenge, traditional CL methods often rely on storing past data for replaying during new tasks Lopez-Paz & Ranzato (2017); Buzzega et al. (2020), which can raise concerns about memory usage and privacy. Besides, prior work shows that replay methods result in overfitting and poor generalization Lopez-Paz & Ranzato (2017); Verwimp et al. (2021); del Rio et al. (2023). To overcome these limitations, recent methods leverage the strong generalization ability of pretrained models (Han et al., 2021; Jia et al., 2022) to solve sequences of CL tasks. A notable line of work is the prompt-based approach (Wang et al., 2022b; Smith et al., 2023; Li et al., 2024b), where a small set of learnable prompts is injected into pretrained backbones for adapting emerging tasks over time.

While these prompt-based methods have demonstrably achieved impressive results, they only consider forgetting caused by changes of common parameters across tasks during learning (Wang et al., 2022c;b) or the inherent mismatch between the chosen prompts at training and testing (Wang et al., 2023; Tran et al., 2023; Zhanxin Gao, 2024). In this work, we further complement these views by *showing that the forgetting of old tasks can potentially come from the uncontrolled overlap between old and new emerging class representations* in the latent space. That is, models will become more confused in distinguishing these classes, resulting in performance degradation w.r.t previous tasks over time (i.e., forgetting). Furthermore, we find that existing methods only *utilize limited information* from the training dataset and often *treat classes equally* during training. Consequently, they overlook the opportunities to enhance the distinguishability of models, especially between the class representations of old and new tasks, and thus, hinder the models' ability to mitigate forgetting.

In addition, we see that human learning behavior has many valuable aspects, especially analyzing data, organizing them in a meaningful way, and finding connections between old and new knowledge (Schön, 1983; Sweller, 1988; Mayer, 2005). Inspired by these practices, we investigate the characteristics of common benchmark datasets, as well as the behavior of pretrained models. Our findings reveal that the data can be categorized into consistent groups, regardless of their arrival times. Each of these groups usually includes classes with similar semantic information that models may confuse and, thus, should be paid more attention to during training. See Appendices C and D.

Therefore, we propose a novel training strategy that constantly arranges emerging class labels in groups, following a *tree-like taxonomy*. In particular, *during training a new task, models are trained to distinguish all classes so far, especially focusing on classes within the same leaf group*. We observe that images belonging to concepts/labels within each of these *leaf groups* share strong visual and semantic correlations, leading to overlap in the latent space, which compromises performance. Thus, by encouraging models to contrast these classes more distinctly, we can reduce the overlap of easily confused ones. This strategy not only mitigates forgetting when new classes emerge, but also consolidates domain-specific knowledge in each leaf group. In addition, based on human learning habits, we find that individuals with stronger foundational knowledge often absorb new information faster and more effectively. Thus, we propose an optimal transport-based technique to further utilize priori from pretrained models, where their initial behaviors will provide another perspective on the relationships between classes.

**Contribution.** We introduce a method named *Reflecting and Linking knowledge: Dynamic Label Structures for Prompt-based Continual Learning* (RefCL). Our main contributions are as follows:

- Inspired from Cognitive Science, we propose a novel approach to reduce forgetting by examining the relationships between data (i.e., *reflecting and linking old and new knowledge*). By dynamically constructing label-based hierarchical taxonomies and leveraging the prior knowledge of pretrained models via an optimal transport-based approach, we can identify the challenging knowledge areas that require further focus during the sequence of tasks.

- Experimentally, our method demonstrated significant superiority over state-of-the-art methods on various continual learning benchmarks.

**Organization.** Firstly, we present related work in Section 2. Next, we formulate the problem and summarize the causes of forgetting in prompt-based Continual Learning models in Section 3. Following that, we discuss the motivation provided by insights from Cognitive Science, and then present the proposed training strategy, emphasizing the importance of leveraging relationships between class data (Section 4). We then present the experimental results to demonstrate the effectiveness of our method (Section 5). Finally, we discuss the limitations and suggest future directions in Section 6.

## 2 RELATED WORK

**Class Incremental Learning (CIL).** This is one of the most challenging and widely studied CL scenarios (Van de Ven & Tolias, 2019; Wang et al., 2023; He et al., 2025), where task identity is unknown during testing, and data of previous tasks is inaccessible during current training. We follow the setting of CIL and propose a novel approach for prompt-based CL models.

**Prompt-based Continual Learning.** This line of work employs the power of pretrained backbone to quickly adapt to the sequence of downstream tasks by updating just a small number of parameters/ prompts for different tasks. In initial work like Wang et al. (2022c;b); Smith et al. (2023), the absence of explicit training constraints often leads to feature overlapping between classes from different tasks. Therefore, recent methods employ some types of contrastive loss (Wang et al., 2023; Li et al., 2023) or utilize Vision Language models (Wang et al., 2022a; Nicolas et al., 2024) to better separate features from tasks. Nevertheless, they treat all classes equally during training, missing the opportunity to efficiently distinguish classes, which have many similarities and are easily confused. Alternatively, we propose a novel approach to reveal the relationships between class data, allowing the model to identify and develop a deeper understanding of the respective knowledge areas, thereby effectively improving model performance.

**Using hierarchical label structures.** In Deep Learning, existing work (Dimitrovski et al., 2011; Chalkidis et al., 2020; Yi et al., 2022; Liu et al., 2021) have considered using hierarchical label systems for efficient representation learning. However, these approaches require all labels in advance, which is not suitable for Continual Learning. Recent researches in CL (Lee et al., 2023; Cao et al.,

2024) have considered hierarchical structures to either expand the label set to create new learning scenarios or manage memory buffers for rehearsal settings. However, these structures cannot indicate the semantic relationships between class labels for effective representation learning. Alternatively, in the line of work where text information is utilized effectively and creatively (Lee et al., 2025; Li et al., 2024a), we introduce a new approach to dynamically building a label-based taxonomy over time for CL models. With the participation of experts, the tree is gradually constructed as new tasks arise, suggesting relationships between class data and enhancing the learning of new knowledge while effectively reducing the forgetting of old information.

**When comparing our label-based taxonomy strategy with the use of VLMs** such as (Lee et al., 2025; Li et al., 2024a; Snæbjarnarson et al., 2025; Liu et al., 2025), we only need to train the vision model with the mapping from the label-based taxonomy, whereas VLMs require training both the vision and language components. Therefore, our approach is more time-efficient.

## 3 BACKGROUND

### 3.1 PROBLEM FORMULATION

We consider the Class Incremental Learning setting (Zhou et al., 2024; Lopez-Paz & Ranzato, 2017; Wang et al., 2023), where a model has to learn from a sequence of $T$ visual classification tasks without revisiting old task data during training or accessing task IDs during inference. Each task $t \in \{1, ..., T\}$ has a respective dataset $\mathcal{D}_t$, containing $n_t$ i.i.d. samples $(\boldsymbol{x}_t^i, y_t^i)_{i=1}^{n_t}$. In this work, we design our model as a composition of two components: *a pretrained Vision Transformer (ViT) (Dosovitskiy et al., 2021) backbone* $f_\Phi$ and *a classification head* $h_\psi$. That is, we have the model parameters $\theta = (\Phi, \psi)$. Similar to other existing prompt-based methods Wang et al. (2023); Smith et al. (2023), we incorporate into the pretrained ViT backbone a set of prompts $\boldsymbol{P}$. We denote the overall network after incorporating the prompts as $f_{\Phi, \boldsymbol{P}}$.

### 3.2 FORGETTING IN PROMPT-BASED METHODS

In Continual Learning, *forgetting* refers to the phenomenon where performance of previous tasks decreases over time. Most prompt-based CL methods, which leverage the power of pretrained models, attribute forgetting either to **(I)** changes in backbone parameters when using the common prompt pool $\boldsymbol{P}$ for all tasks (Wang et al., 2022c;b) or to **(II)** the inherent mismatch between the models used at training and testing, as discussed and analyzed in Zhanxin Gao (2024); Tran et al. (2023). Other work such as (Li et al., 2023) suggests that the cause also arises from **(III)** *overlapping between old and new emerging class representations, which bear high semantic resemblance to previous samples*. In Appendix B, we also provide empirical studies to complement the third views, confirming that the overlap causes confusion in distinguishing between old and new classes, thereby reducing the performance of learned tasks over time. This is coincidentally similar to the behavior of human memory, where although the old knowledge exists somewhere in the brain, still be confused due to the newly acquired information (Anderson & Neely, 1996; Wimber et al., 2015; Loftus, 2005; Nader et al., 2000; Wixted, 2004). This motivates us to propose a novel method, *focusing on identifying easily confused class pairs*, thereby reducing forgetting and improving model performance.

## 4 PROPOSED METHOD

In the previous section, we noted that increasing overlap in data representations as more tasks arrive is one of the main reasons leading to performance degradation (i.e., forgetting). Therefore, *if we can identify groups of easily confused classes/concepts, we will better enhance the distinguishability of models and thus reduce forgetting*. In addition, insights from Cognitive Science (Section 4.1), regarding the benefits of organizing information in a meaningful way, suggest us to arrange labels from CL tasks in a hierarchical taxonomy to identify these groups of classes.

Interestingly, we find that we can build the tree where concepts within the same leaf group tend to be visually and semantically similar. These concepts likely cause more overlap in the visual latent space and causing confusion for the models (See Appendix D). Motivated by this observation, we propose group-based contrastive learning in Section 4.2 to maximize the separability of these concepts, thus mitigating forgetting over time. Furthermore, to reinforce the method's effectiveness, Section 4.3 presents an optimal transport-based technique, harnessing priori of pretrained backbones from a new perspective.

## 4.1 MOTIVATION

**Insights from Cognitive Science.** Research in Cognitive Science highlights the importance of *reflection, and organization of information* as critical components for effective learning. Studies show that when learners take time to reflect on their experiences, they deepen their understanding and enhance retention (Schön, 1983). This reflective practice encourages individuals to connect new information with existing knowledge, fostering a more integrated learning experience (Bransford et al., 2000). Moreover, organizing information into coherent structures, such as outlines or concept maps, allows learners to see relationships between concepts, making it easier to retrieve and reflect the information later (Mayer, 2005).

**Our Approach.** It is evident that besides *reflection* - comparison of old and new knowledge, the key factor in learning efficiently is *to organize and link information in an insightful way*, where concepts are arranged according to their semantic meanings. This observation motivates us *to develop deep learning models that learn concepts structured in hierarchical taxonomies.*

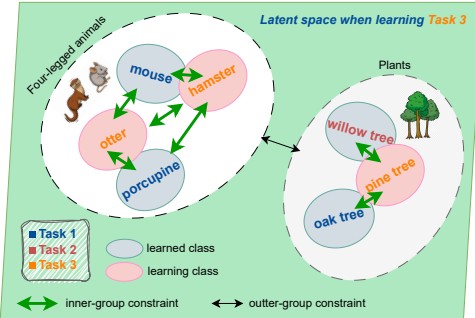

Figure 1: ***Problem:*** When new classes arrive (e.g., learning Task 3), the latent space of a model for all tasks so far becomes fuller, and class representations tend to be overlapped, leading to performance degradation in old tasks. ***Our solution:*** We focus more on separating easily confused classes (using *"inner-group constraint"* - green arrows ↔) whose concepts/labels lie in the same leaf group on a label-based hierarchical taxonomy, suggested by expert knowledge. The *"outer-group constraint"* briefly represents for the common constraints like Cross Entropy Loss/ Supervised Contrastive loss, applying for all class so far. *[Best viewed in color mode]*

Specifically, we propose *structuring data labels* in tree-like taxonomies, which can be flexibly and consistently expanded over time using domain expertise. Based on this structure, we have a view of the relationships between all classes so far, especially identifying which classes belong to the same group with many shared characteristics, easily confused, and require more focus to distinguish.

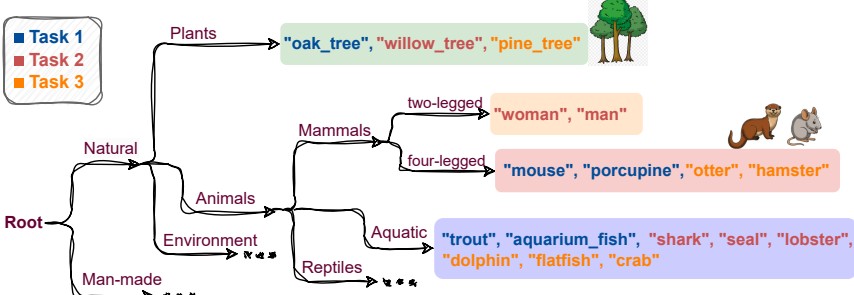

Figure 2: **The label-based hierarchical taxonomy** when learning Task 3, on Split-CIFAR100. The colors (i.e., **blue**, **red**, **orange**) of the label names represent the task order in which the corresponding classes appear. Accordingly, the tree-like taxonomy is gradually developed and detailed over time. *[Best viewed in color mode]*

Taking the learning process of Split-CIFAR-100 as an example, when training on task $t = 3$, we can construct a tree-like taxonomy of concepts/labels, as shown in Figure 2. We observe that the classes under a same leaf in the structure (e.g., oak tree, willow tree, and pine tree in the leaf group "plants") exhibit strong visual and semantic correlations. Consequently, as shown in Figure 3a, these class features are more overlapping with each other rather than with another class from other leaf groups (e.g., "four-legged"). In this way, the tree-like taxonomy serves as a tool to help identify easily confused classes, facilitating the subsequent process of making them more separable in the feature space (See Appendix D to see the alignment between our label-based taxonomies and visual latent space). Leveraging the insight obtained from the taxonomy above, we aim to train a network so that all class representations must be distinct, *especially those within each leaf group*. In this way, the overlap between classes—particularly between old and new ones—is significantly reduced, which effectively mitigates forgetting and enhances model performance.

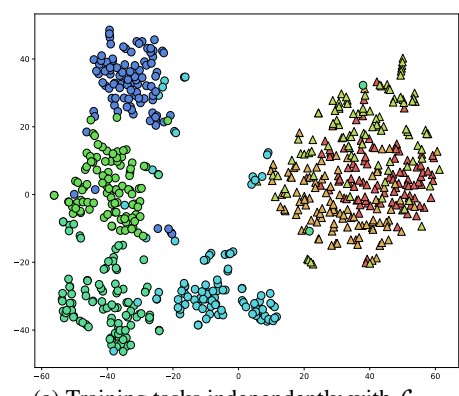 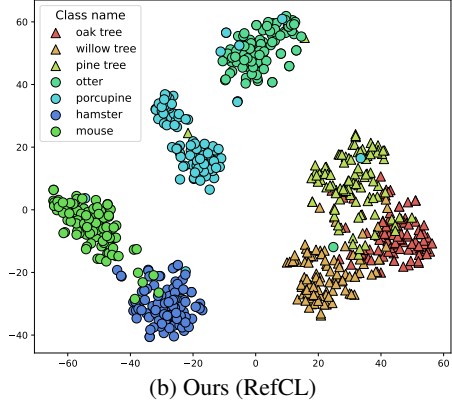

(a) Training tasks independently with $\mathcal{L}_{CE}$     (b) Ours (RefCL)

Figure 3: **t-SNE visualizations** of classes within leaf groups of Four-legged animals (● circular points) and Plants (▲ triangular points) when learning Task 3, Split-CIFAR-100. The appearance order of the classes: Task 1 - *"mouse", "porcupine", "oak tree"*; Task 2 - *"willow tree"*; Task 3 - *"otter", "hamster", "pine tree"* (also refer to Fig. 1 and 2). We can see that if we train tasks independently with $\mathcal{L}_{CE}$ as in existing work like (Wang et al., 2022c; Smith et al., 2023), the classes within each leaf group, which arrive at different time, can be overlapped seriously (Fig. 3a). Alternatively, by using the our taxonomy-based strategy, we can effectively *reflecting and linking* old and new knowledge, thus enhancing model representation learning and reducing forgetting (Fig. 3b).

## 4.2 Leveraging Hierarchical Taxonomies for Better Representation Learning

During the training process, whenever a new class appears, its label name is automatically added to the tree-like taxonomy, into a leaf group containing classes with similar characteristics (Figure 2). To develop this hierarchical structure, we can rely on expert knowledge, which can help incrementally construct a meaningful related tree (refer to Appendix C for further details). Structuring information in this way will *provides useful insights during training, indicating how each knowledge is related to the other and which group of concepts requires further focus.*

Particularly, for each task $t$, we dedicate a set of prompt $\boldsymbol{P}_t$. We aim to minimize overlap between all classes so far, especially focusing on increasing the separability between classes belonging to the same leaf group on the taxonomy (e.g., four-legged mammals, plants, etc.,). That is, *taxonomy acts as a reference information channel to support the training process.* Let $g$ be a leaf group on the taxonomy $\mathcal{G}$ (i.e., $g \in \mathcal{G}$), $X_k^g$ and $Y_k^g$ denote the corresponding sets of input samples and labels under the group $g$ that belong to the task $k$ ($k \leq t$). Besides Cross Entropy loss $\mathcal{L}_{CE}$, we propose using a regularization loss function for sample each $\boldsymbol{x}$ that arrives in task $t$ and belongs to leaf group $g$ as follows:

$$\mathcal{L}_{\mathcal{G}}(\psi, \boldsymbol{P}_t, \boldsymbol{x}) = \alpha \mathcal{L}_g(\psi, \boldsymbol{P}_t, \boldsymbol{x}) + \beta \mathcal{L}_{all}(\psi, \boldsymbol{P}_t, \boldsymbol{x}) \tag{1}$$

where we have defined

$$\mathcal{L}_g(\cdot) = -\log \sum_{\boldsymbol{x}' \in X_{1\dots t}^g | y_{\boldsymbol{x}'} = y_{\boldsymbol{x}}} \frac{u(\boldsymbol{z}_x \cdot \boldsymbol{z}_{x'})}{\sum_{\bar{x} \in X_{1\dots t}^g} u(\boldsymbol{z}_x \cdot \boldsymbol{z}_{\bar{x}})},$$

$$\mathcal{L}_{all}(\cdot) = -\log \sum_{\boldsymbol{x}' \in X_{1\dots t} | y_{\boldsymbol{x}'} = y_{\boldsymbol{x}}} \frac{u(\boldsymbol{z}_x \cdot \boldsymbol{z}_{x'})}{\sum_{\bar{x} \in X_{1\dots t}} u(\boldsymbol{z}_x \cdot \boldsymbol{z}_{\bar{x}})}$$

are the Supervised Contrastive losses that we put on class representation *within leaf group $g$*, and *all classes* so far, respectively; and $u(\boldsymbol{z}_x \cdot \boldsymbol{z}_{x'}) = \exp(\frac{\boldsymbol{z}_x \cdot \boldsymbol{z}_{x'}}{\tau})$, with $\boldsymbol{z}_x = f_{\Phi, \boldsymbol{P}_t}(\boldsymbol{x})$ is the feature vector on the latent space of the prompt-based model and $y_{\boldsymbol{x}}$ is the ground truth label of $\boldsymbol{x}$, $\tau$ is the temperature with $\tau = 0.1$ for all experiments, and $\alpha$ is the coefficient that controls how much we want to force classes within leaf group $g$ stay apart further. [1]

---

[1]Note that because the data of old tasks can not be accessed when training the new one, we follow previous work (Wang et al., 2023; Li et al., 2024b) to encode the information of each trained class $c$ in the form of a Gaussian Mixture model $\text{GMM}_c = \{\mathcal{N}(\mu_{c,i}, \Sigma_{c,i})\}_{i=1}^K$, at the end of the corresponding task, where $K$ is the number of components. Then in task $t$, each representation $\boldsymbol{z}_{\bar{x}}, \bar{x} \in X_c$ of each old class $c$, is sampled from $\text{GMM}_c$, to compute loss functions in Eq. (1).

Looking closer at Eq. (1), this equation implies that when learning a new task, the model will be encouraged to identify the decision boundary between all old and new classes (i.e., using $\mathcal{L}_{all}$ - outer-group constraint). Besides, it will especially focusing on those in the same *leaf group*, which often share many common characteristics and can confuse classification models (i.e., using $\mathcal{L}_g$ - inner-group constraint), see Figure 1.

**Discussion.** This manipulation matches the way our brain naturally works: Without reflecting and linking, we would be confused about old and new concepts, especially those sharing many common characteristics, leading to incorrect judgments and decisions in practice. For AI models, learning new tasks without thoroughly considering the learned information of old tasks can lead to uncontrolled overlapping in the latent space, resulting in forgetting the learned knowledge and harming final performance. In this work, thanks to the insight from hierarchical taxonomies, we can identify which group of classes that easy to get overlapped in the latent space (see classes "oak tree, pine tree, willow tree" in Figure 3a), thereby actively intervening and enhancing the model representation learning, thus reducing forgetting in the CL environment.

### 4.3 HARNESSING PRIOR KNOWLEDGE FROM PRETRAINED MODELS VIA OPTIMAL TRANSPORT

Considering a leaf group, there may be data classes with varying levels of overlap in the latent space. For example, Figure 4 provides statistical results regarding the relative position of classes in leaf groups "plants" and "four-legged mammals" in the latent space of a pretrained model. We can see that in the group of animals, the L2-Wasserstein distance between class representations of "mouse" and "porcupine" (30.69) is more significant than the one between "mouse" and "hamster" (26.05), meaning that the class representation of the second pair can overlap more than the first one. This empirical result suggests that although the strategy in Section 4.2 helps improve the model's ability to recognize difficult-to-identify classes, we still treat all classes in that group equally. Thus, *the algorithm may inadvertently ignore the pairs of classes that are easily confused and need to be further distinguished.*

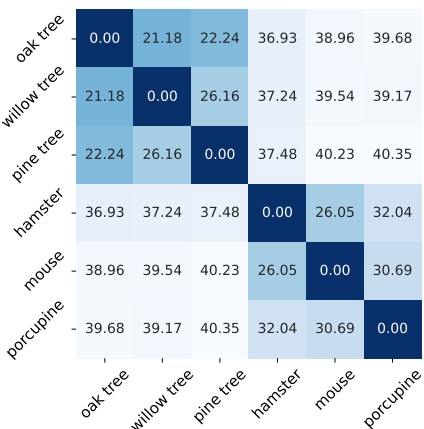

Figure 4: L2-Wassertein distance between classes (Split-CIFAR-100) in latent space of a pretrained backbone (Sup-21K). ***Within a leaf group*** (i.e., "plant" or "four-legged mammals"), ***there may be data classes with varying levels of correlation*** in the latent space.

Additionally, pretrained models are extensively trained on large datasets, which results in their substantial generalization abilities. That is the reason why they are considered as good starting points for the adaptation to downstream tasks (Devlin et al., 2019; Brown et al., 2020; He et al., 2016; Raffel et al., 2020). Therefore, we propose utilizing the pretrained models from another novel perspective, which can comprehend the use of the label-based taxonomy in Section 4.2 during training. Particularly, we take advantage of these pretrained models to obtain prior assumptions about the relationships between the class representations, thereby elegantly intervening in better separating each pair of classes within each leaf group.

More specifically, to extract the relationship between the classes, we compute the L2-Wasserstein distance ($W_2$) between feature distributions of each class pair. We denote $D_{c_i}^{\Phi}$ be the distribution of class $c_i$ on the latent space of the pretrained model $f_{\Phi}$ - to distinguish it from the one on prompted latent space of $f_{\Phi,\boldsymbol{P}}$, as described in Section 4.2. The distribution $D_{c_i}^{\Phi}$ is obtained in the form of a Gaussian Mixture model at the end of the respective task, *only once and then kept fixed for the next reuse; therefore, it does not impose a computational burden* (See Appendix E.3). In this way, when $t$ tasks have arrived, we have the corresponding sets of distributions $\{D_c^{\Phi}\}_{c \in Y_{1,t}}$ of all $m_t$ classes so far. Thus, we can gradually complete the WD-based matrix $M$, showing the distance between pairs of classes as follows:

$$M \in \mathbb{R}^{m_t \times m_t}, \text{ where } M_{i,j} = W_2(D_{c_i}^{\Phi}, D_{c_j}^{\Phi}). \tag{2}$$

The fact is that the smaller the $W_2$ between the latent distribution of two classes, the more overlapped they are, and the more force may be needed to separate them. Thus, we employ these results to obtain the corresponding weight factors, which control how much constraint needs to be applied to each class pair. Particularly, we compute *the weight matrix*:

$$\Gamma = [\boldsymbol{\gamma_{ij}}]_{m_t \times m_t} = [1/\exp(M_{ij}/\delta)]_{m_t \times m_t} \tag{3}$$

where $\delta$ is a temperature. We then apply this information to obtain a weighted version of $\mathcal{L}_g$, in which the closer the two class distributions are, the larger the weight assigned, and they will be further focused to push away:

$$\mathcal{L}'_g(\cdot) = -\log \sum_{\boldsymbol{x}' \in X^g_{1..t} | y_{\boldsymbol{x}'} = y_{\boldsymbol{x}}} \frac{u(\boldsymbol{z}_x \cdot \boldsymbol{z}_{x'})}{\sum_{\bar{x} \in X^g_{1..t}} \boldsymbol{\gamma_{y_x y_{\bar{x}}}} u(\boldsymbol{z}_x \cdot \boldsymbol{z}_{\bar{x}})} \tag{4}$$

This strategy is completely economical and aligns well with the CL learning scheme, as the distance between representations of each class pair only needs to be computed once, and the matrix $M$ is continuously expanded when new classes arrive. Practically, when learning a new task, the first epoch is for capturing information about the behavior of the pretrained model on the data of this task. Moreover, this approach is also aligned with the findings in Cognitive Science (Osgood & Bower, 1953; Baltes, 1987), showing that the accumulated experiences of past learning (i.e., knowledge contained in pretrained models) create momentum for future learning (i.e., adapting model for a sequence of downstream tasks).

Finally, the final objective function of our full method can be formulated as follows:

$$\mathcal{L} = \mathcal{L}_{CE} + \mathcal{L}_{\mathcal{G}}, \text{ where } \mathcal{L}_{\mathcal{G}} = \alpha \mathcal{L}'_g + \beta \mathcal{L}_{all} \tag{5}$$

## 5 EXPERIMENT

### 5.1 EXPERIMENTAL SETUP

**Datasets.** We use 4 common CIL benchmarks, including Split CIFAR-100, Split ImageNet-R, 5-Datasets, and Split CUB-200.

**Baselines.** We compare ours with 8 typical and state-of-the-art prompted-based methods, including L2P (Wang et al., 2022c), DualPrompt (Wang et al., 2022b), CODA-Prompt (Smith et al., 2023), HiDe-Prompt (Wang et al., 2023), OVOR (Huang et al., 2024), ITA-$IA^3$ (Porrello et al., 2025), APT (Chen et al., 2025), RainbowPrompt (Hong et al., 2025).

**Metrics.** We use two metrics: Final Average Accuracy (FAA) and Final Forgetting Measure (FFM). Please refer to *Appendix A* for further details, including datasets, baselines, metrics, and other training configurations.

### 5.2 EXPERIMENTAL RESULT

**Our approach achieves superior results compared to baselines.** Table 1 presents the overall performance comparison between our proposed method and other baselines. The key observation is that our method is the strongest one, because the gap compared with the runner-up method being up to around 1.5% of FAA on all the datasets. Additionally, our method avoids forgetting better than all baselines, by a gap up to 17% on 5-Datasets.

**Ablation studies.** Figure 5 reports the ablation studies of our training strategy. Particularly, compared to training tasks independently using Cross Entropy loss $\mathcal{L}_{CE}$ like in DualP and L2P, exploiting the relationships between data with the label-based taxonomy and the OT-based strategy (ours) helps improve FAA by about 5% to 10% (Figure 5a). Besides, when examining the role of exploiting additional prior information from pretrained backbones using the OT approach upon the taxonomy, we see that FAA is improved from 0.6% to 0.8% (Figure 5b). These results demonstrate the positive impact of this component. In both figures, the improvements on Split-CIFAR100 and 5-Datasets are the lowest, while it is more pronounced on Split-CUB-200. This may be because the groups of these two datasets (Split-CIFAR100 and 5-Datasets) have fewer overlapping classes, as the classes in each group likely have more recognizable features. Meanwhile, Split-CUB-200 is a dataset of birds with images that can be difficult for human eyes to recognize, thus so our method performs

Table 1: **Overall performance comparison.** We provide FAA and FFM of all methods, with standard deviation taken over at least 3 runs of different random seeds. The results corresponding to the best FAA among baselines are underlined.

| Method | Split CIFAR-100 | | Split ImageNet-R | | 5-Datasets | | Split CUB-200 | |
|---|---|---|---|---|---|---|---|---|
| | **FAA** (↑) | FFM (↓) | **FAA** (↑) | FFM (↓) | **FAA** (↑) | FFM (↓) | **FAA** (↑) | FFM (↓) |
| L2P | 83.06 ±0.17 | 6.58 ±0.40 | 63.65 ±0.12 | 7.51 ±0.17 | 81.84 ±0.95 | 4.58 ±0.53 | 74.52 ±0.92 | 11.25 ±0.23 |
| DualPrompt | 86.60 ±0.19 | 4.45 ±0.16 | 68.79 ±0.31 | 4.49 ±0.14 | 77.91±0.45 | 13.17 ±0.71 | 82.05±0.95 | 3.56 ±0.53 |
| OVOR | 86.68 ±0.22 | 5.25 ±0.12 | 75.72 ±0.82 | 5.77 ±0.12 | 82.34 ±0.48 | 4.83 ±0.35 | 78.12 ±0.65 | 8.13 ±0.52 |
| CODA-Prompt | 86.94 ±0.63 | 4.04 ±0.18 | 70.03 ±0.47 | 5.17 ±0.22 | 64.20 ±0.53 | 17.22 ±0.55 | 74.34 ±0.68 | 12.05 ±0.41 |
| HiDe-Prompt | 92.61 ±0.28 | 1.52 ±0.10 | 75.06 ±0.12 | 4.05 ±0.19 | 93.92 ±0.33 | 2.31 ±0.12 | 86.62 ±0.35 | 2.55 ±0.15 |
| ITA-$IA^3$ | 91.08±0.31 | 2.25 ±0.18 | 72.75±0.26 | 5.65 ±0.31 | 85.75±0.45 | 3.62±0.28 | 84.23±0.42 | 3.83±0.38 |
| APT | 89.22±0.65 | 3.21 ±0.52 | 79.40±0.47 | 4.38 ±0.46 | 80.25 ±0.56 | 4.26±0.43 | 78.52±0.95 | 7.65 ±0.88 |
| Rainbow-Prompt | 89.86±0.11 | 3.44 ±0.26 | 79.09±0.13 | 3.90 ±0.23 | 81.23±0.45 | 4.02±0.52 | 84.65±0.42 | 3.84 ±0.35 |
| Ours (RefCL) | **93.94** ±0.23 | **1.05** ±0.15 | **79.65** ±0.12 | **3.06** ±0.22 | **95.02** ±0.20 | **1.21** ±0.15 | **87.93** ±0.22 | **2.01** ±0.23 |

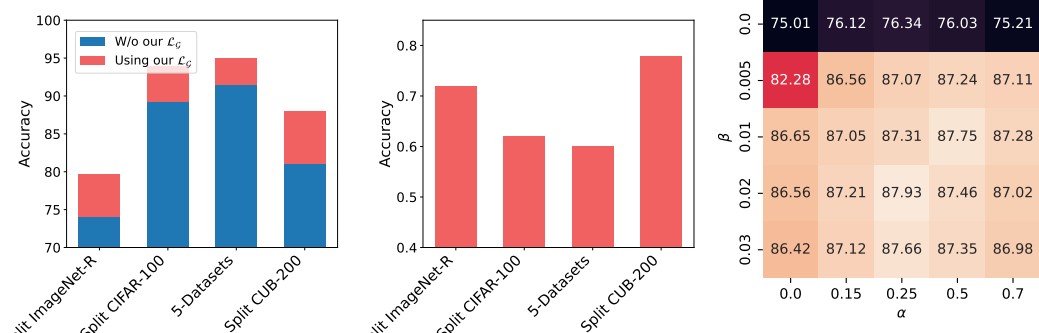

(a) The role of our $\mathcal{L}_\mathcal{G}$ - Eq (1)    (b) The role of OT technique in $\mathcal{L}_\mathcal{G}$   (c) Varying the value of $\alpha, \beta$ (Split-CUB-200)

Figure 5: **Ablation studies** about our training strategy. Figure *(b)* depicts the performance improvement after applying our OT-based technique to our taxonomy strategy (Eq.4). The efficiency of this technique *w/o our taxonomy strategy* is further analysed in ***Appendix E.4***.

better. Furthermore, Figure 5c provides the experimental results when varying $\alpha$ and $\beta$. The data show significant degradations of model performance when each loss function of $\mathcal{L}_\mathcal{G}$ is eliminated, demonstrating their specific roles. In particular, w/o using our taxonomy and OT-based strategy ($\alpha\mathcal{L}_g = 0$), the performance can decrease up to 5.65%. Besides, the effect of $\mathcal{L}_{all}$ can be clearly seen when $\beta = 0$, as it helps constrain the relative correlations of all classes.

**t-SNE visualization on latent space.** Figure 3 illustrates the effect of our method in improving the model's representation learning on Split-CIFAR100. Specifically, the classes are better clustered, and the separation between them is more distinct. Especially, the classes "oak tree", "willow tree", and "pine tree" are divided into clear clusters, rather than being mixed together as in the traditional training strategy, where tasks are trained independently. We also provide other visualizations on other datasets in Appendix E.2.

**The impact of $\delta$ in OT technique.** Figure 6 illustrates the dependence of model performance on the value of the coefficient $\delta$ in our OT technique (see Eq.3). The results show that if $\delta$ is too small, the performance can go down significantly, even worse than HiDE-Prompt. This occurs because the values of *the weight matrix* will be too large, leading to classes, which have strong correlations in the latent

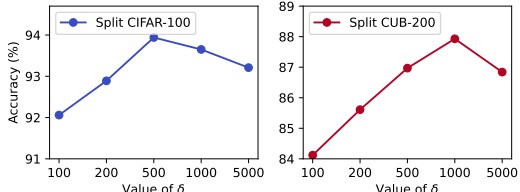

Figure 6: Model performance when $\delta$, in Eq (3), varies.

space of $f_\Phi$, being pushed too far apart. This causes an imbalance compared to the overall correlation of all classes and unexpected overlapping. Conversely, if $\delta$ is set too large, the corresponding weight will be small, preventing achieving optimal efficiency. For Split-CIFAR-100 and Split-CUB-200, the optimal values are $\delta = 500$ and $100$, respectively.

**The alignment between the label-based taxonomy and visual space**   The trees used in our experiment are built based on the information of the labels and the visual identity of the corresponding data (See Appendix C). The success of our experiment comes from the appropriate correlation between the visual space and the arrangement of their labels on the taxonomy. In the main paper, we present Figure 2, a part of the tree obtained when training Split-CIFAR100, along with the tSNE visualization in Figure 3a, and the distributional distance between pairs of classes in Figure 4. The results show that basically, if labels are on the same leaf group, the visual space will likely share a lot of common information and have a high possibility of overlapping, causing the model to be confused and make wrong judgments. We provide in Figure 7 other measures of average cosine similarity and Euclidean distance between pairs of classes, the results basically show the consistency and explain why our method works. Similar results for other datasets are provided in Appendix D.

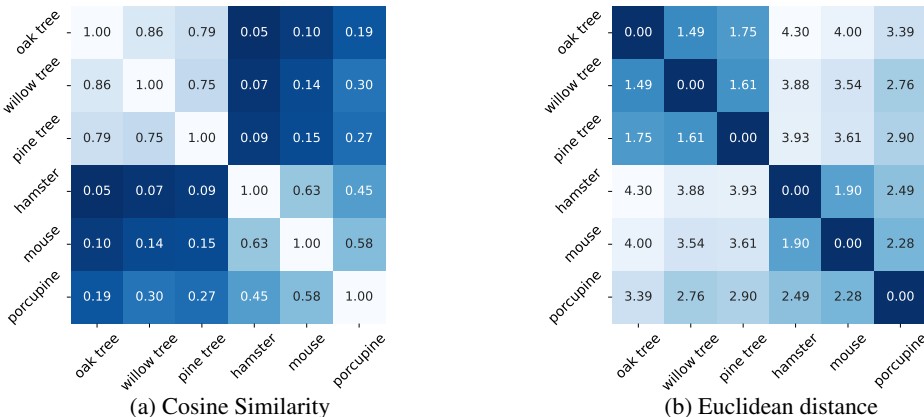

(a) Cosine Similarity                          (b) Euclidean distance

Figure 7: Correlation in latent space of class pairs (Split-CIFAR-100).

**The superiority of our proposed method on various types of pretrained backbones.**   Table 2 demonstrates that our method consistently outperforms the strongest baseline, HiDE-Prompt across all cases. This confirms the superior effectiveness of our method across a diverse range of types of pretrained models.

Table 2: Comparison when using different pretrained backbones.

| Backbone | Split CIFAR-100 | | Split Imagenet-R | | 5-Datasets | | Split CUB-200 | |
|---|---|---|---|---|---|---|---|---|
| | RefCL | HiDE-Prompt | RefCL | HiDE-Prompt | RefCL | HiDE-Prompt | RefCL | HiDE-Prompt |
| Sup-21K | **93.94** | 92.61 | **79.65** | 75.06 | **94.96** | 93.83 | **87.92** | 86.56 |
| iBOT-21K | **94.01** | 93.02 | **75.12** | 70.83 | **95.21** | 94.88 | **80.06** | 78.23 |
| iBOT-1K | **94.27** | 93.48 | **75.80** | 71.33 | **94.59** | 93.89 | **79.22** | 78.54 |
| DINO-1K | **94.12** | 93.51 | **72.25** | 68.11 | **94.20** | 93.50 | **78.98** | 78.42 |
| MoCo-1K | **92.32** | 91.57 | **68.23** | 63.77 | **94.22** | 93.28 | **78.32** | 77.63 |

**Other results.**   We also provide experimental results in ***Appendix E***, related to model performance when using different taxonomy structures, and analyze other aspects of our OT technique, the coefficients $\alpha, \beta, \tau$ when using loss functions. Additionally, we verify the correlation between taxonomy and visual space when considering other datasets and pretrained models, as well as discuss the current limitations and potential impacts of the proposed method.

## 6   CONCLUSION

In this work, we demonstrate the importance of meaningfully organizing data information rather than just lumping them together for training. Particularly, we propose arranging data labels into tree-like taxonomies to identify data groups that are likely to confuse models. This approach encourages the models to focus and develop deeper knowledge about each group, reducing forgetting and motivating more effective learning in subsequent tasks. Additionally, we propose leveraging the initial behavior of pretrained models to obtain hidden structures of training data, providing a new perspective to further enhance performance. Finally, the experimental results demonstrate our effectiveness.

**Limitations.**   Despite this novel approach, the quality of the hierarchical taxonomy depends on the quality of expert knowledge. For example, if similar image classes are not assigned to the same leaf group in this label-based taxonomy, the constraint we put on each such group may not perform as expected.

## REPRODUCIBILITY STATEMENT

In order to facilitate the reproduction of our empirical results, we provide detailed descriptions of the experimental setup in Section 5 and Appendix A. All datasets used in this study are publicly available, enabling full replication of our experiments.

## USE OF LARGE LANGUAGE MODELS

In accordance with the ICLR 2026 policy, we disclose our use of Large Language Models (LLMs) during the preparation of this paper. Large language models were employed for *(i)* editorial purposes, including grammar correction and spelling refinement. (ii) The generation of label-based taxonomies, which help guide the training process of the main model, as described in Appendix C.

However, all scientific ideas, model design, and experimental results reported in this paper are entirely conceived and executed by the authors. The LLM was never used to generate research concepts, hypotheses, or experimental findings. The authors take full responsibility for the content of the paper.

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

# Supplement to "Reflexing and Linking knowledge: Dynamic Label Structures for Prompt-based Continual Learning"

## A EXPERIMENTAL SETTINGS

### A.1 DATASETS

We adopt the following common benchmarks:

- **Split CIFAR-100** (Krizhevsky et al., 2009): This dataset includes images from 100 different classes, each being relatively small in size. The classes are randomly organized into 10 sequential tasks, with each task containing a unique set of classes.
- **Split ImageNet-R** (Krizhevsky et al., 2009): This dataset contains images from 200 extensive classes. It includes difficult examples from the original **ImageNet** dataset, as well as newly acquired images that display a variety of styles. The classes are randomly divided into 10 distinct incremental tasks.
- **5-Datasets** (Ebrahimi et al., 2020): This composite dataset incorporates **CIFAR-10** (Krizhevsky et al., 2009), **MNIST** (LeCun et al., 1998), **Fashion-MNIST** (Xiao et al., 2017), **SVHN** (Netzer et al., 2011), and **notMNIST** (Bulatov, 2011). Each of these is treated as a separate incremental task, enabling the evaluation of the impact of substantial variations between tasks.
- **Split CUB-200** (Wah et al., 2011): This dataset contains fine-grained images of 200 distinct bird species. It is randomly divided into 10 incremental tasks, each with a unique subset of classes.

### A.2 BASELINES

In the main paper, we use CL methods with pretrained ViT as the backbone. We group them into (a) the group using a common prompt pool for all tasks, and (b) the group dedicating distinct prompt sets for each task:

(1) **L2P** (Wang et al., 2022c): The first prompt-based work for continual learning (CL) suggested using a common prompt pool, selecting the top $k$ most suitable prompts for each sample during training and testing. This approach might facilitate knowledge transfer between tasks but also risks catastrophic forgetting. Unlike our approach, L2P doesn't focus on training classifiers or setting constraints on features from old and new tasks during training, which may limit the model's predictability.

(2) **DualPrompt** (Wang et al., 2022b): The prompt-based method aims to address L2P's limitations by attaching complementary prompts to the pretrained backbone, rather than only at input. DualP introduces additional prompt sets for each task to leverage task-specific instructions alongside invariant information from the common pool. However, like L2P, it does not focus on efficiently learning the classification head. Additionally, selecting the wrong prompt ID for task-specific instructions during testing can negatively impact model performance.

(3) **OVOR** (Huang et al., 2024): while using only a common prompt pool for all tasks, this work introduces a regularization method for Class-incremental learning that uses virtual outliers to tighten decision boundaries, reducing confusion between classes from different tasks. Experimental results demonstrate the role of representation learning, which focuses on reducing overlapping between class representations.

(4) **CODA-Prompt** (Smith et al., 2023): This prompt-based approach uses task-specific learnable prompts for each task. Similar to L2P, CODA employs a pool of prompts and keys, computing a weighted sum from these prompts to generate the real prompt. The weights are based on the cosine similarity between queries and keys. To avoid task prediction at the end of the task sequence, the weighted sum always considers all prompts. CODA improves over DualP and L2P by optimizing keys and prompts simultaneously, but it still hasn't addressed the drawbacks mentioned for DualP.

(5) **HiDe-Prompt** (Wang et al., 2023): a recent SOTA prompt-based method that decomposes learning CIL into 3 modules: a task inference, a within-task predictor and a task-adaptive predictor. The

second module trains prompts for each task with a contrastive regularization that tries to push features of new tasks away from prototypes of old ones. To predict task identity, it trains a classification head on top of the pretrained ViT. TAP is similar to a fine-tuning step that aims to alleviate classifier bias using the Gaussian distribution of all classes seen so far. However, this method does not declare the relationship between data during training, thereby missing the opportunity to improve model performance.

(6) **ITA-**$IA^3$ (Porrello et al., 2025): Provides theoretical analysis and valuable insights related to demystify compositionality in standard non-linear networks through the second order Taylor approximation of the loss function. The proposed formulation highlights the importance of staying within the pre-training basin to achieve composable modules. Moreover, it provides the basis for two dual incremental training algorithms: the one from the perspective of multiple models trained individually, while the other aims to optimize the composed model as a whole. In our experiment, we compare our method with the first version in this paper.

(7) **APT** (Chen et al., 2025): This method proposed training a unified set of shared prompts for all tasks and, instead of concatenating these prompts to the input, directly alters the attention computation of the CLS token by incorporating the prompts. This straightforward and lightweight design significantly lowers computational complexity—reducing both inference costs and the number of trainable parameters—while also removing the necessity to optimize prompt lengths for various downstream tasks. This results in a more efficient and effective solution for rehearsal-free class-incremental learning.

(8) **Rainbow-Prompt** (Hong et al., 2025): Different from other methods, this paper introduced an innovative prompt-evolving mechanism that adaptively combines base prompts (i.e., task-specific prompts) into a single unified prompt while maintaining diversity. By transforming and aligning both previously learned and newly introduced base prompts, model will continuously updates accumulated knowledge to support the learning of new tasks. Additionally, they proposed to use a learnable probabilistic gate that dynamically decides which layers to activate during the evolution process.

### A.3 METRICS

In our study, we employed two key metrics: the Final Average Accuracy (FAA) and the Final Forgetting Measure (FFM). To define these, we first consider the accuracy on the $i$-th task after the model has been trained up to the $t$-th task, denoted as $A_{i,t}$. The average accuracy of all tasks observed up to the $t$-th task is calculated as $AA_t = \frac{1}{t}\sum_{i=1}^{t} A_{i,t}$. Upon the completion of all $T$ tasks, we report the Final Average Accuracy as FAA $= AA_T$. Additionally, we calculate the Final Forgetting Measure, defined as FFM $= \frac{1}{T-1}\sum_{i=1}^{T-1} \max_{t\in\{1,\dots,T-1\}}(A_{i,t} - A_{i,T})$. The FAA serves as the principal indicator for assessing the ultimate performance in continual learning models, while the FFM evaluates the extent of catastrophic forgetting experienced by the model.

### A.4 IMPLEMENTATION DETAILS

Our implementation basically aligns with the methodologies employed in prior research Wang et al. (2022c); Smith et al. (2023); Chen et al. (2025). Specifically, our framework incorporates the use of a pretrained Vision Transformer (ViT-B/16) as the backbone architecture. For the optimization process, we utilized the Adam optimizer, configured with hyper-parameters $\beta_1$ set to 0.9 and $\beta_2$ set to 0.999. The training process was conducted using batches of 24 samples, and a fixed learning rate of 0.03 was applied across all models except for CODA-Prompt. For CODA-Prompt, we employed a cosine decaying learning rate strategy, starting at 0.001. Additionally, a grid search technique was implemented to determine the most appropriate number of epochs for effective training. Regarding the pre-processing of input data, images were resized to a standard dimension of $224 \times 224$ pixels and normalized within a range of $[0, 1]$ to ensure consistency in input data format.

In Table 1 of the main paper, the results of L2P, DualPrompt, CODA-Prompt, and HiDe-Prompt on Split CIFAR-100 and Split ImageNet-R are taken from (Wang et al., 2023). Their results on the other two datasets are produced from the official code provided by the authors. For the remaining baselines, the reported results are also reproduced from their official code.

## B  FORGETTING IN PROMPT-BASED CONTINUAL LEARNING METHODS - ANALYSE THE CAUSE W.R.T EMERGENCE OF NEW DATA

In Continual Learning, *forgetting* refers to the phenomenon in which performance on previously learned tasks decreases over time. Current prompt-based CL methods, which leverage the power of pretrained models, attribute forgetting either to **(I)** changes in backbone parameters when using the common prompt pool $P$ for all tasks (Wang et al., 2022c;b) or to **(II)** the inherent mismatch between the models used at training and testing, as discussed and analyzed in Zhanxin Gao (2024); Tran et al. (2023). Other work such as (Li et al., 2023) suggests that the cause also arises from **(III)** *overlapping between old and new emerging class representations, which bear high semantic resemblance to previous samples*. In this section, we also provide empirical studies to complement the third views:

| Dataset | Feature shift |
|---|---|
| Split-Imagenet-R | 0 |
| Split-CIFAR-100 | 0 |
| Split-CUB-200 | 0 |

(a) Feature shift corresponding to $\mathcal{D}_1$ after learning the sequence of tasks.

(b) *"Within task accuracy"* on $\mathcal{D}_1$.

(c) *Model accuracy* on $\mathcal{D}_1$

Figure 8: **Empirical study about forgetting.** We setup the experiment to eliminate factors *(I)* and *(II)*, which result in feature shift after learning the sequence of tasks - **Table (a)**. Therefore, the *"within task accuracy"* on $\mathcal{D}_1$, using classification head $s_1(\boldsymbol{x})$ to classify classes within task 1 only, remains over time - **Figure (b)**. However, when using head $h_\psi$ to classify all classes observed so far, the *model accuracy* on $\mathcal{D}_1$ decrease significantly - **Figure (c)**, suggesting that besides *feature shift, there are other factors that lead to forgetting*.

To uncover the third factor causing forgetting, we deliberately consider the cases where the behavior of backbone w.r.t each learned task is unchanged over time - meaning (I) and (II) would not happen. Firstly, to eliminate concerns about changing learned parameters (I), we consider methods that propose using a distinct set of prompts $P_t$ to a specific task $t$. Thus, we conduct experiments on HiDE-Prompt (Wang et al., 2023), which is the latest SOTA in prompt-based CL. Then, the remaining potential factor is the difference between the prompt chosen at inference time. To this end, the training is carried out as usual; when testing, we intentionally choose the right prompt for each sample to ensure that there is no change in the backbone's behavior compared to training. And the features after backbone will be classified normally without taskID.

For a detailed examination, we propose an experimental setup to analyze the performance degradation of corresponding models over time. Specifically, we train a classification head $s_1(\boldsymbol{x})$ on the latent space $f_{\Phi, P_1}(\boldsymbol{x})$ derived from the first task $\mathcal{D}_1$. As the model undergoes continual training on a sequence of subsequent tasks, we assess the ability of $s_1(\boldsymbol{x})$ to classify instances from $\mathcal{D}_1$ over time, defining this as *within-task accuracy*. This is different from *model accuracy on* $\mathcal{D}_1$, which is obtained when we use the usual classification head $h_\psi$ as the original design of HiDE-Prompt. This classification head $h_\psi$ considers all classes that the model has observed so far, instead of just data of $\mathcal{D}_1$ as $s_1(\boldsymbol{x})$. The results in Figure 8b, show that *within-task accuracy* definitely remains. In contrast, Figure 8c, illustrates a significant decrease in *true accuracy*, raising the question of whether we have overlooked additional factors contributing to final forgetfulness.

Additionally, considering the *inference feature space* of $f_{\Phi, P}(\boldsymbol{x})$, we can see that as more tasks arrive, the number of classes grows up, increasing the possibility of overlap between class distributions. To support this point, we illustrate the representations of some class images in Figure 1, as well as the corresponding t-SNE visualizations in Figure 3. In particular, after Task 1, we have representations of "oak tree", "mouse", and "porcupine" located in quite separate locations. However, when Task 2 and then Task 3 arrive, the appearance of classes "willow tree" and "pine tree" makes

class "oak tree" no longer maintain the separation from the remaining classes as before, *leading to a remarkable drop in performance of previous tasks*.

Based on the evidence above, we can see that the overlap causes confusion in distinguishing between old and new classes, thereby reducing the performance of learned tasks over time (i.e., forgetting). This is coincidentally similar to the behavior of human memory, where although the old knowledge exists somewhere in the brain, still be confused due to the newly acquired information (Anderson & Neely, 1996; Wimber et al., 2015; Loftus, 2005; Nader et al., 2000; Wixted, 2004). Recognizing this motivates us to propose a novel method, focusing on identifying easily confused class pairs, thereby reducing forgetting and improving performance.

## C USING LLMs (I.E., CHATGPT, LLAMA, ETC.,) TO BUILD TREE-LIKE TAXONOMY DURING A SEQUENCE OF TASKS, INCREMENTALLY

**How to generate taxonomy for training?** To support the training process, we need the help of experts to arrange the labels in the appropriate positions on the tree, thereby achieving the desired effects during the training process, helping the model focus more on difficult knowledge areas. In our experiments, we use LLMs as an expert simulation tool. In particular, we use the following prompt structure to generate the taxonomies:

```
Given the label list: ['···'], provide me the taxonomy from this
list, based on their origin, type, and shape, so that the image
encoders can recognize their images.
```

Example output, when the list ['···'] is ["leopard", "rabbit", "mouse", "camel", "trout", "aquarium_fish", "snake", "rose", "lawn_mower", "bottle"]:

```
taxonomy = {
    "Natural": {
        "Animals": {
            "Mammals": {
                "Four-legged": ["leopard",
                "rabbit", "mouse", "camel"]
            },
            "Aquatic": ["trout",
            "aquarium_fish"],
            "Reptiles": ["snake"]
        },
        "Plants": {
            "Flowers": ["rose"]
        }
    },
    "Man-Made": {
        "Objects": {
            "Tools": ["lawn_mower"],
            "Containers": ["bottle"]
        }
    }
}
```

Below is an example of generated taxonomies for each task of Split-CIFAR100:

```
T1 = {
    "Natural": {
        "Animals": {
            "Mammals": {
                "Four-legged": ["leopard",
                "rabbit", "mouse", "camel"]
            },
```

```
              "Aquatic": ["trout",
              "aquarium_fish"],
              "Reptiles": ["snake"]
          },
          "Plants": {
              "Flowers": ["rose"]
          }
      },
      "Man-Made": {
          "Objects": {
              "Tools": ["lawn_mower"],
              "Containers": ["bottle"]
          }
      }
}

T2 = {
    "Natural": {
        "Animals": {
            "Mammals": {
                "Four-legged": ["leopard",
                "rabbit", "mouse", "camel",
                "otter"]
            },
            "Aquatic": ["trout",
            "aquarium_fish",
                        "shark", "seal",
                        "lobster"],
            "Reptiles": ["snake"]
        },
        "Plants": {
            "Flowers": ["rose", "tulip"],
            "Trees": ["palm_tree"]
        }
    },
    "Man-Made": {
        "Objects": {
            "Tools": ["lawn_mower"],
            "Containers": ["bottle", "bowl"]
        },
        "Vehicles": {
            "Wheeled": ["motorcycle"]
        },
        "Structures": {
            "Buildings": ["skyscraper",
                          "house"]
        }
    }
}

T3 = ...
}
```

The taxonomies for other datasets are available in our source code.

**Regarding the issue of CIL constraint violation?**   The model adheres to the training principles of Class-Incremental Learning (CIL) since it inherently lacks any foresight into future data. Our model only accesses data available up to the current training stage, along with the corresponding taxonomy

established by the expert. While the expert contributes to the training process, this support is limited to information relevant to the data that the model has already observed.

**Flexibility of taxonomy** *(i)* Label-based taxonomies can be built with any labeled dataset. The architecture of these taxonomies can be diverse (as illustrated in Figures 12 and 13), depending on the expert's knowledge. However, it is necessary to ensure that the layers on the leaf node are arranged properly for the model to be trained effectively, which comes from the expert's knowledge. *(ii)* In our experiment, these label-based taxonomies are continuously and consistently expanded as new tasks emerge, mirroring how human understanding grows as we acquire new knowledge, reflect on it, and connect it to what we already know.

## D    VERIFICATION OF THE ALIGNMENT BETWEEN LABEL-BASED TAXONOMIES AND VISUAL LATENT SPACE

The label-based trees/taxonomies used in our experiment are built based on the information of the labels and the visual identity of the corresponding data (See Appendix C). The success of our experiment comes from the appropriate correlation between the visual space and the arrangement of their labels on the taxonomy. In the main paper, we present Figure 2, a part of the tree obtained when training Split-CIFAR100, along with the t-SNE visualization in Figure 3a, and the distributional distance between pairs of classes in Figure 4. The results show that basically, if labels are on the same leaf group, the visual space will likely share a lot of common information and have a high possibility of overlapping, causing the model to be confused and make wrong judgments. We also provide in Figure 7 other measures of average cosine similarity and Euclidean distance between pairs of classes, the results basically show the consistency and explain why our method works.

To better demonstrate the intuition that there is a correlation between the label-based tree-like taxonomy and visual space, we provide visualization results corresponding to different datasets in Figure 9, 10, 11. The results show that classes within the same *leaf group* often share many common visual characteristics and likely have strong correlation and close distance in the latent space. Therefore, identifying these such groups on the taxonomies will provide us with a reference channel to determine easily confusable classes, thereby enhancing the models' representation learning.

## E    ADDITIONAL EXPERIMENTS

### E.1    HOW DO DIFFERENT HIERARCHICAL TAXONOMY STRUCTURES AFFECT MODEL PERFORMANCE?

Table 3: Performance comparison when using different LLMs to generate the corresponding taxonomy

| LLMs used to build taxonomies | Split CIFAR-100 | | Split Imagenet-R | | Split CUB-200 | |
|---|---|---|---|---|---|---|
| | Sup-21K | iBOT-21K | Sup-21K | iBOT-21K | Sup-21K | iBOT-21K |
| *Baseline (HiDE-Prompt)* | *92.61* | *93.02* | *75.06* | *70.83* | *86.56* | *78.23* |
| Llama-3-70b-Groq | 93.58 | 93.82 | **80.01** | **75.12** | **88.00** | **79.05** |
| GPT-4o-Mini | **93.94** | **94.01** | 79.65 | 74.12 | 87.93 | 79.02 |
| Gemini-1.5-Pro | 93.35 | 93.43 | 78.76 | 73.42 | 87.02 | 78.43 |
| DeepSeek-R1 | 93.56 | 93.62 | 79.24 | 73.93 | 87.65 | 78.72 |
| Mistral-Medium | 93.12 | 93.35 | 78.54 | 73.12 | 87.00 | 78.22 |
| Claude-3.5-Haiku | 93.65 | 93.62 | 79.12 | 73.98 | 87.83 | 78.75 |
| Average results of LLMs | $93.53_{\pm 0.28}$ | $93.64_{\pm 0.24}$ | $79.18_{\pm 0.52}$ | $73.95_{\pm 0.69}$ | $87.57_{\pm 0.45}$ | $78.8_{\pm 0.33}$ |

Table 3 shows the performance of models when using the most advanced LLMs to support the process of building label-based hierarchical taxonomies. Although the output samples presented in Figures 12 and 13 indicate some notable differences in the approaches as well as the final results

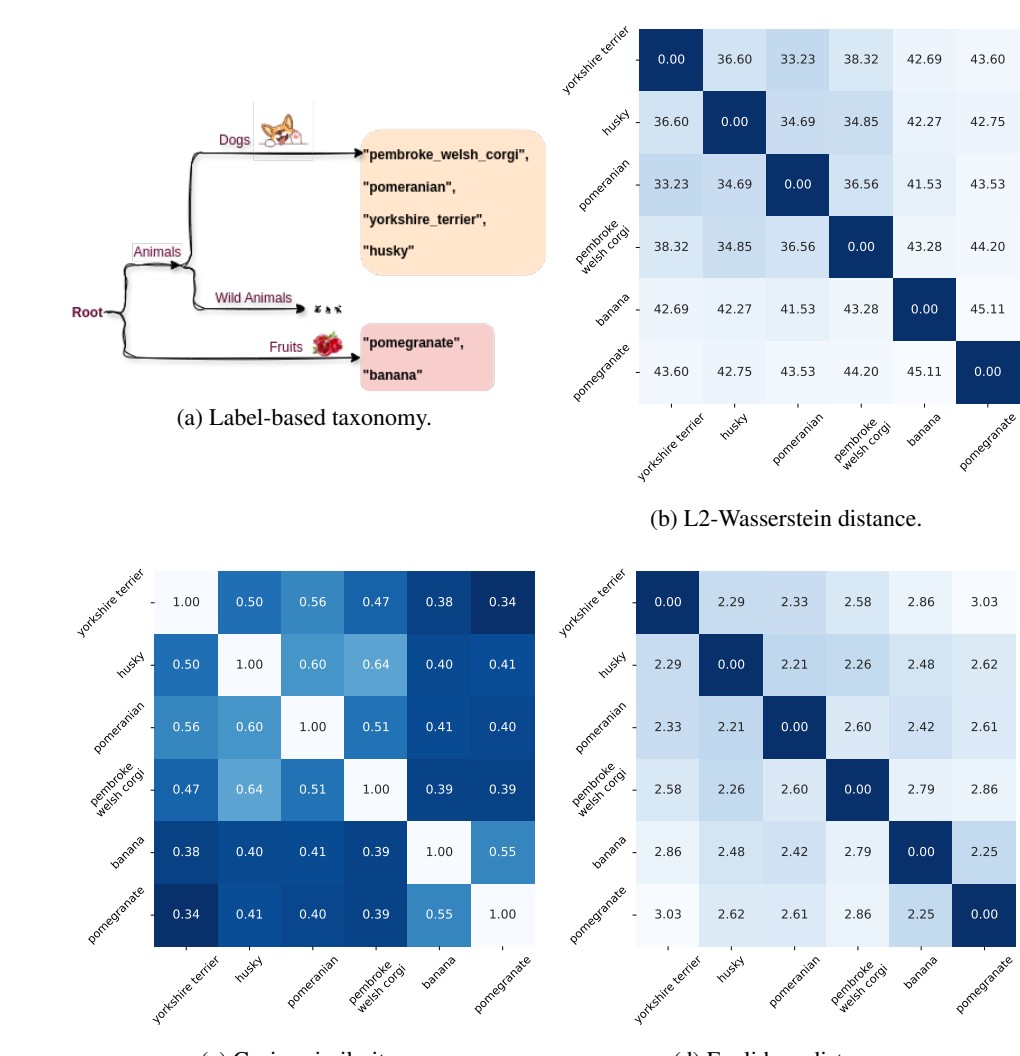

(a) Label-based taxonomy.

(b) L2-Wasserstein distance.

(c) Cosine similarity.

(d) Euclidean distance.

Figure 9: Relationship between visual features and taxonomies (Split-Imagenet-R)

of the corresponding LLMs' trees constructed, the numerical results show that there is not much difference when using these different LLMs.

This may stem from the sufficiently good knowledge and semantic capabilities of these powerful models. That is, despite certain differences, most class labels are appropriately organized into their corresponding leaf groups. Furthermore, the involvement of our OT-based technique once again helps to modify the constraint level of the taxonomy-based strategy, resulting in not much difference when using these LLMs. Overall, Llama-3-70B-Groq and GPT-4o-Mini are the two models that yield the best results.

## E.2 T-SNE VISUALIZATION ON DIFFERENT DATASETS

Figures 14, 15, 16 present visualizations of the latent space, demonstrating the effectiveness of our method in actively linking knowledge and focusing on difficult knowledge areas, instead of learning tasks independently like many current methods.

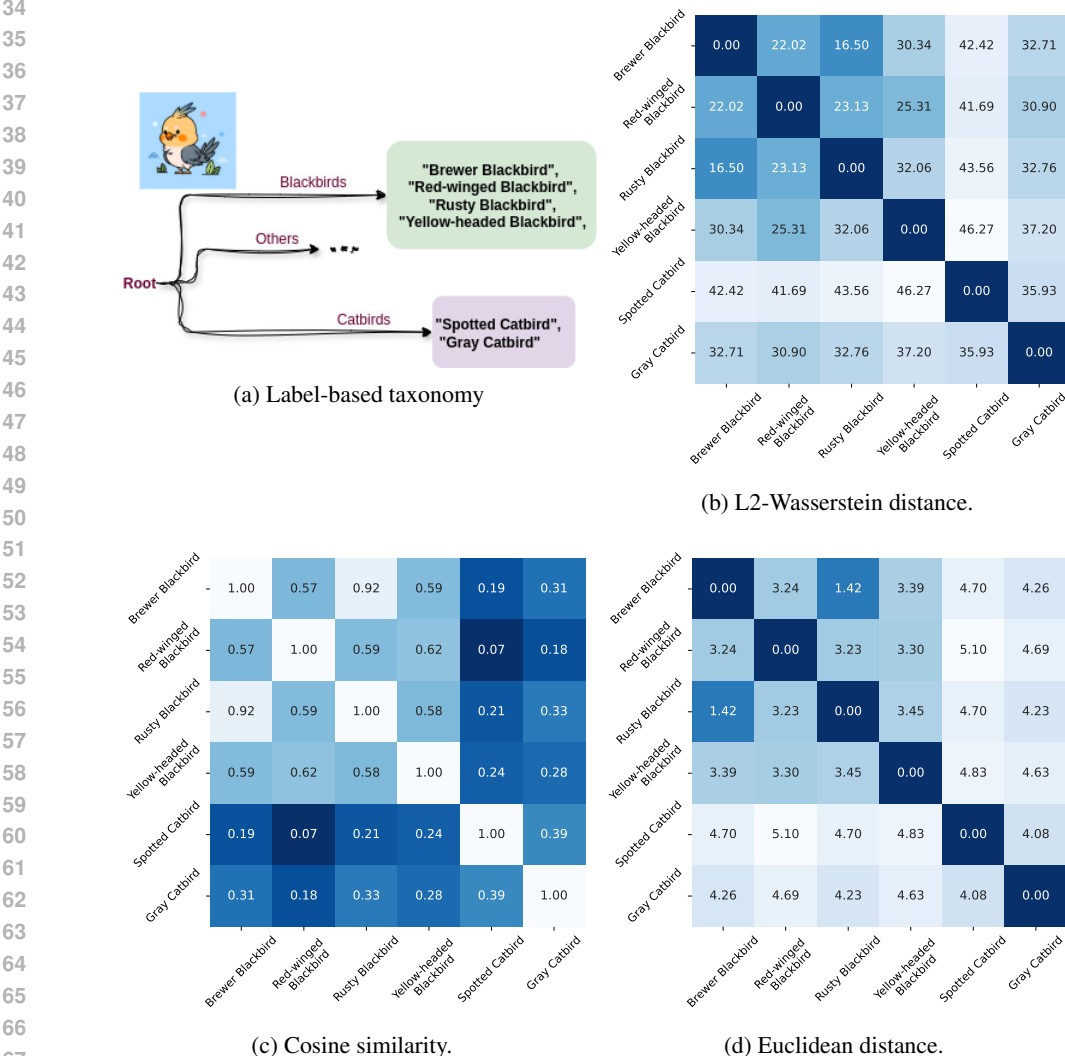

(a) Label-based taxonomy

(b) L2-Wasserstein distance.

(c) Cosine similarity.

(d) Euclidean distance.

Figure 10: Relationship between visual features and taxonomies (Split-CUB-200)

### E.3 Computational cost for computing Wasserstein distance-based matrix in out OT-based strategy (Section 4.3)

The results in Table 4 show that the computational cost of our Wasserstein distance matrix does not cause computational burden during training, and does not affect the testing process.

### E.4 The effectiveness of our OT-based technique

This section provides an additional perspective on the effectiveness of our OT-based technique (Section 4.3). Instead of demonstrating the effectiveness of this technique in reinforcing the strength of the taxonomy strategy as mentioned in the main paper, we conduct experiments to show its effectiveness when applied to conventional supervised contrastive loss. That is, at this point, the model's objective function is as follows:

$$\mathcal{L} = \mathcal{L}_{CE} + \mathcal{L}_{\mathcal{G}} = \mathcal{L}_{CE} + \mathcal{L}'_{all} \tag{6}$$

where $\mathcal{L}'_{all}$ is the new version of $\mathcal{L}_{all}$ after applying our OT-based strategy:

$$\mathcal{L}'_{all}(\cdot) = -\log \sum_{\boldsymbol{x}' \in X_{1\ldots t} | y_{\boldsymbol{x}'} = y_{\boldsymbol{x}}} \frac{u(\boldsymbol{z}_x \cdot \boldsymbol{z}_{x'})}{\sum_{\bar{x} \in X_{1\ldots t}} \boldsymbol{\gamma}_{\boldsymbol{y}_{\boldsymbol{x}} \boldsymbol{y}_{\bar{\boldsymbol{x}}}} u(\boldsymbol{z}_x \cdot \boldsymbol{z}_{\bar{x}})} \tag{7}$$

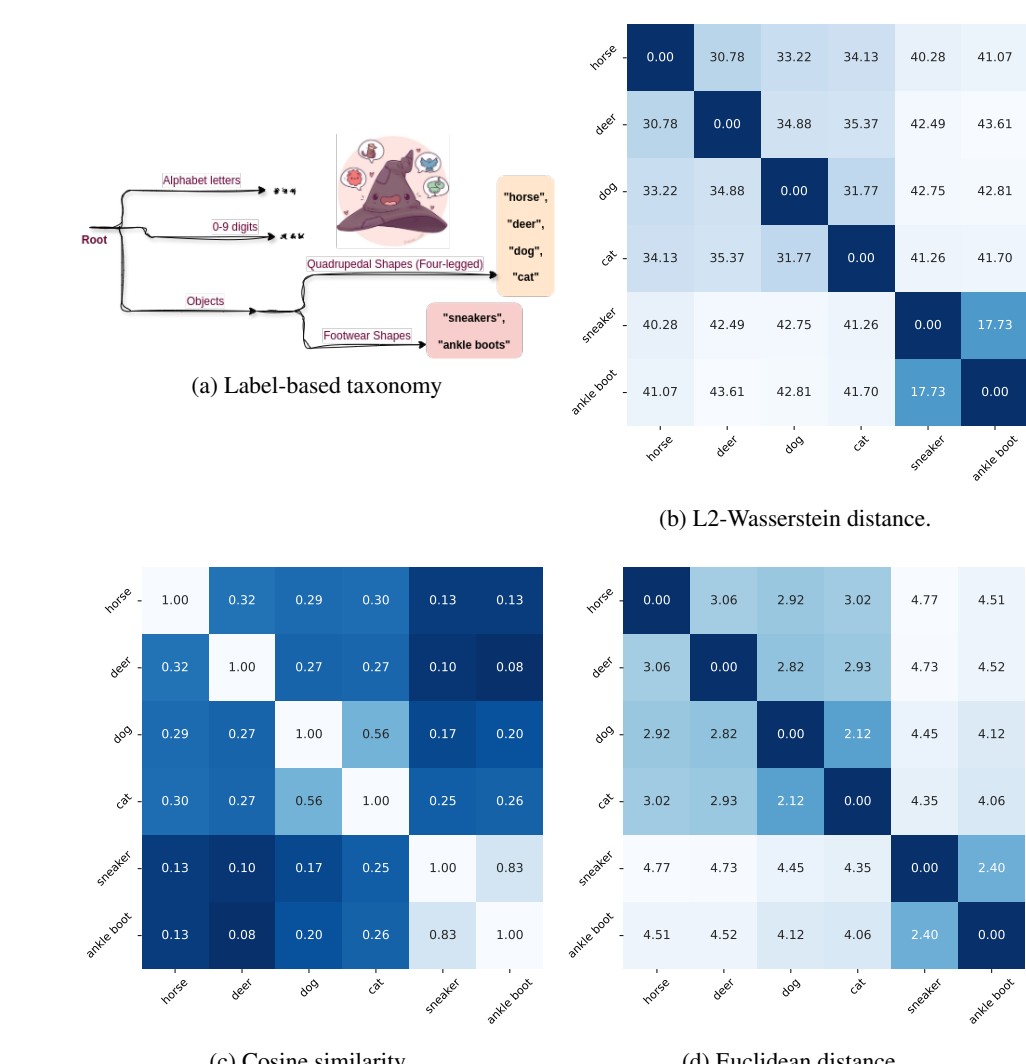

(a) Label-based taxonomy

(b) L2-Wasserstein distance.

(c) Cosine similarity.

(d) Euclidean distance.

Figure 11: Relationship between visual features and taxonomies (5-Datasets)

Table 4: Computational cost

| Training cost (min/task) | Split-CIFAR-100 | Split-ImageNet-R | Split-CUB-200 | 5-Datasets |
|---|---|---|---|---|
| HiDE-Prompt | 42.83 | 81.61 | 17.52 | 155.42 |
| Ours | 43.92 | 83.02 | 28.15 | 157.15 |
| Ours (WS computation only) | 0.024 | 0.095 | 0.095 | 0.006 |

| WD Computation (S-CIFAR100) (s) | Until Task 1 | Until Task 5 | Until Task 10 | |
|---|---|---|---|---|
| ViT-B16 | 0.015 | 0.36 | 1.43 | |
| Vit-L16 | 0.015 | 0.36 | 1.43 | |
| *(num of classes so far)* | 10 | 50 | 100 | |

| Inference time (ms/sample) | S-CIFAR100 | S-ImageNetR | S-CUB200 | 5-Datasets |
|---|---|---|---|---|
| HiDE-Prompt | 21.92 | 21.92 | 21.92 | 21.92 |
| Ours | 21.92 | 21.92 | 21.92 | 21.92 |

Based on this, we conducted the related experiments and presented the results in Table 5. The data show that the performance of the model when using the objective function in Eq.6 *(B)* is nearly

```
T2 = {
    "Natural": {
        "Animals": {
            "Mammals": {
                "Four-legged": ["leopard", "rabbit", "mouse", "camel", "otter"]},
            "Aquatic": ["trout", "aquarium_fish", "shark", "seal", "lobster"],
            "Reptiles": ["snake"]},
        "Plants": {
            "Flowers": ["rose", "tulip"],
            "Trees": ["palm_tree"]}
    },
    "Man-Made": {
        "Objects": {
            "Tools": ["lawn_mower"],
            "Containers": ["bottle", "bowl"]},
        "Vehicles": {
            "Wheeled": ["motorcycle"]},
        "Structures": {
            "Buildings": ["skyscraper", "house"]}
    }
}
```

(a) GPT-4o-Mini

```
T2 = {
    "Biological Entities": {
        "Mammals": {
            "Terrestrial Predators": ["Leopard"],
            "Small Terrestrial Mammals": ["Rabbit","Mouse", "Camel"],
            "Aquatic/Semi-Aquatic Mammals": ["Otter", "Seal"]},
        "Marine Life": {
            "Fish": ["Trout", "Aquarium Fish", "Shark"],
            "Crustaceans": ["Lobster"]},
        "Reptiles": ["Snake"]
    },
    "Botanical Entities": {
        "Flowering Plants": ["Rose, Tulip"],
        "Tree": ["Palm Tree"]},
    "Man-Made Objects": {
        "Machinery": ["Lawn Mower", "Motorcycle"],
        "Containers": ["Bottle", "Bowl"],
        "Architectural Structures": ["Skyscraper", "House"]}
}
```

(b) Claude-3.5-Haiku

```
T2 = {
    "Animals": {
        "Mammals": {
            "Terrestrial": ["leopard", "rabbit", "mouse", "camel"],
            "Semi-aquatic": ["otter"],
            "Marine": ["seal"]},
        "Fish": {
            "Freshwater": ["trout", "aquarium_fish"],
            "Saltwater": ["shark"]},
        "Invertebrates": {
            "Crustaceans": ["lobster"]},
        "Reptiles": ["snake"]
    },
    "Plants": {
        "Flowers": ["rose", "tulip"],
        "Trees": ["palm_tree"]},
    "Objects": {
        "Man-made": {
            "Vehicles": ["motorcycle"],
            "Structures": ["skyscraper", "house"],
            "Containers": ["bottle", "bowl"],
            "Tools": ["lawn_mower"]}
    }
}
```

ollama run llama3

(c) Llama-3-70b-Groq

Figure 12: Taxonomy samples when using different LLMs (Llama, GPT, Claude), in *Task 2*, with the label list `["leopard", "rabbit", "mouse", "camel", "trout", "aquarium_fish", "snake", "rose", "lawn_mower", "bottle"]`

equivalent to that of using our taxonomy strategy w / without the OT technique *(C)*, helping to improve performance by almost 6% compared to training tasks independently *(A)*. The reason that *(B)* has not surpassed *(C)* may be because *(B)* is only leveraging prior information from vision pretrained models, while *(C)* also incorporates insights from LLMs regarding the relationships between data classes. The effectiveness of this collaboration is demonstrated when our fully proposed method *(D)* delivers superior results.

```
T2 = {
    "Natural": {
        "Animals": {
            "Mammals": {
                "Felidae": ["leopard"], "Lagomorpha": ["rabbit"],
                "Rodentia": ["mouse"], "Artiodactyla": ["camel"]},
            "Fish": {
                "Freshwater": ["trout", "aquarium_fish"]},
                "Saltwater": ["shark"]},
            "Marine Mammals": ["seal"],
            "Semi-aquatic Mammals": ["otter"],
            "Invertebrates": {
                "Crustaceans": ["lobster"], "Reptiles": ["snake"]}
        },
        "Plants": {
            "Flowers": ["rose", "tulip"], "Trees": ["palm_tree"]}
    },
    "Man-Made": {
        "Containers": ["Bottle", "Bowl"],
        "Structures": ["Skyscraper", "House"]},
        "Vehicles": ["Motorcycle"], "Tools": ["Lawn Mower"]
    }
}
```

(a) Gemini-1.5-Pro

```
T2 = {
    "Biological Entities": {
        "Terrestrial Mammals": ["leopard", "rabbit", "mouse", "camel"],
        "Aquatic/Semi-Aquatic Life": {
            "Freshwater": ["trout", "aquarium_fish"], "Saltwater": ["shark"],
"Amphibious": ["otter", "seal"]},
        "Reptiles/Invertebrates": ["snake", "lobster"],
        "Plants": {
            "Floral": ["rose", "tulip"], "Arboreal": ["palm_tree"]}
    },
    "Man-Made Objects": {
        "Containers": {
            "Vertical": ["bottle"], "Open": ["bowl"]},
        "Structures": {
            "Vertical": ["skyscraper"], "Horizontal": ["house"]},
        "Mechanical Devices": {
            "Vehicle": ["motorcycle"], "Tool": ["lawn_mower"]}
    }
}
```

(b) DeepSeek-R1

```
T2 = {
    "Biological Entities": {
        "Animals": {
            "Mammals": {
                "Terrestrial": {"Quadruped": {
                        "Carnivores": ["leopard"], "Lagomorphs": ["rabbit"],
                        "Rodents": ["mouse"], "Artiodactyla": ["camel"]}},
                "Semi-aquatic": ["otter"],
                "Marine": ["seal"]},
            "Fish": {
                "Freshwater":["trout","aquarium_fish"],"Saltwater": ["shark"]},
            "Invertebrates": {
                "Crustaceans": ["lobster"], "Reptiles": ["snake"]},
        },
        "Plants": {"Flowers": ["rose", "tulip"], "Trees": ["palm_tree"]}
    },
    "Man-Made Objects": {
        "Tools": ["lawn_mower"], "Containers": ["bottle", "bowl"],
        "Vehicles": ["motorcycle"],
        "Structures": ["skyscraper", "house"]}
}
```

(c) Mistral-Medium

Figure 13: Taxonomy samples when using different LLMs (Gemini, DeepSeek, Mistral), in *Task 2*, with the label list `["leopard", "rabbit", "mouse", "camel", "trout", "aquarium_fish", "snake", "rose", "lawn_mower", "bottle"]`

### E.5 EFFECT OF THE TEMPERATURE $\tau$

In each of our experiments, we used a common temperature value $\tau$ for the two supervised contrastive loss functions (i.e., $\mathcal{L}_g$ and $\mathcal{L}_{all}$). Table 6 below shows the model performance when this value varies on Split-CIFAR-100 and Split-Imagenet-R. The results indicate that, in general, $\tau = 0.1$ is the optimal value achieved.

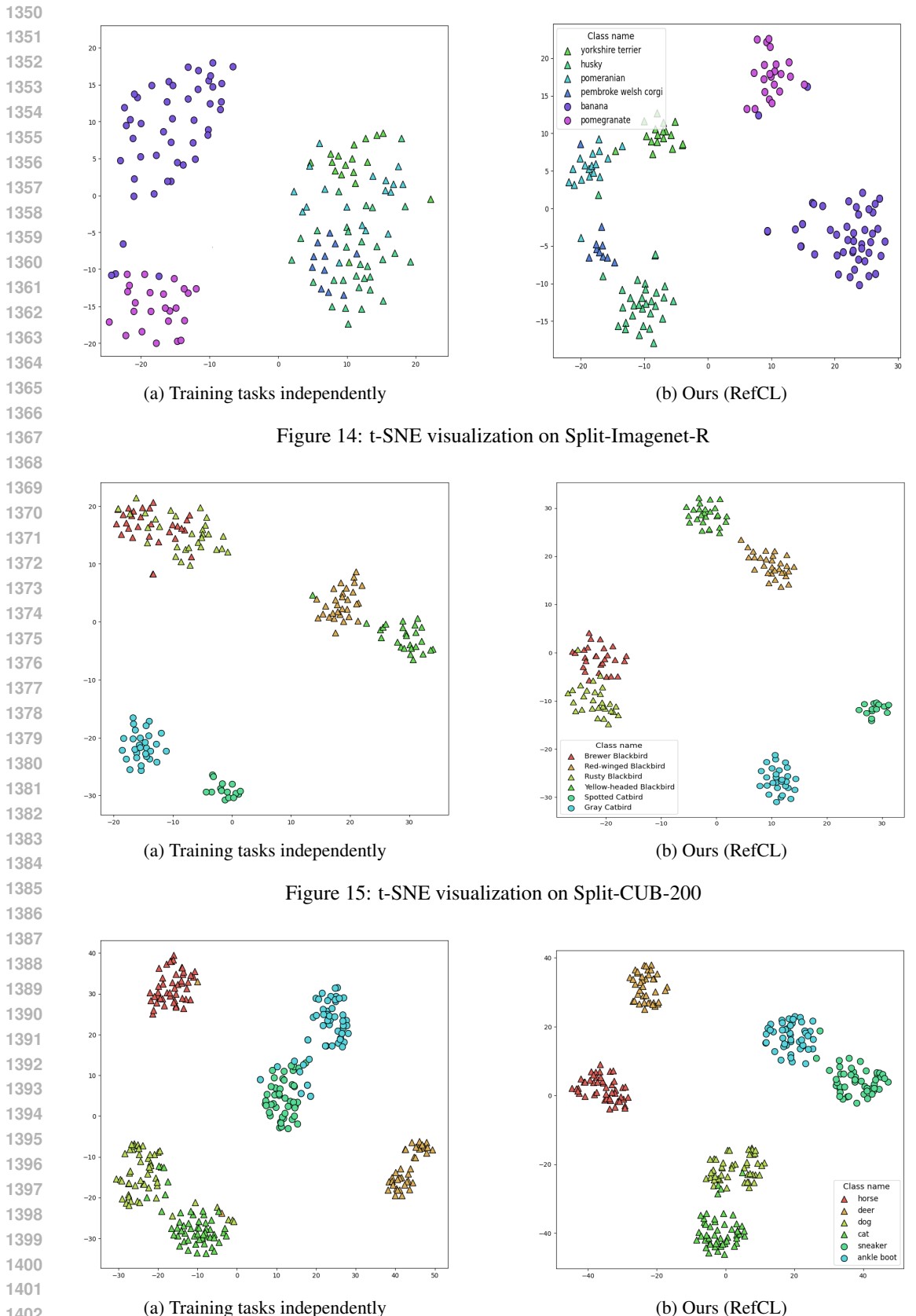

Figure 14: t-SNE visualization on Split-Imagenet-R

Figure 15: t-SNE visualization on Split-CUB-200

Figure 16: t-SNE visualization on 5-Datasets

Table 5: The efficiency of our OT-based technique

| Dataset | Split-CIFAR-100 | Split-ImageNet-R | 5-Dataset | Split-CUB-200 |
|---|---|---|---|---|
| (A) Baseline (training tasks independently, $\mathcal{L}_{CE}$ only) | 87.52 | 73.55 | 91.48 | 81.02 |
| *(B) Using OT-trick w/o taxonomies* | *93.04* | *78.69* | *94.12* | *86.85* |
| (C) Using taxonomy w/o OT-based strategy | 93.31 | 78.72 | 94.42 | 87.18 |
| (D) Ours (both taxonomy and OT-based strategy) | **93.94** | **79.65** | **95.02** | **87.93** |

Table 6: Performance when the temperature factor $\tau$ varies

| Dataset | 0.05 | 0.1 | 0.3 | 0.5 | 0.7 |
|---|---|---|---|---|---|
| Split CIFAR-100 | 90.04 | **93.94** | 91.42 | 90.15 | 88.81 |
| Split ImageNet-R | 77.98 | **79.65** | 78.22 | 76.01 | 74.96 |

### E.6 MODEL PERFORMANCE WHEN $\alpha, \beta$ VARY

Figures 17a and 17b show the changes in model performance when varying the values of $\alpha$ and $\beta$ in the objective function of the proposed method (Equation 5). Accordingly, the roles of $\mathcal{L}'_g$ and $\mathcal{L}_{all}$ are clearly demonstrated and consistent with the insights obtained from the Split-CUB-200 dataset mentioned earlier in the main paper. When either of these quantities is omitted, model performance decreases significantly.

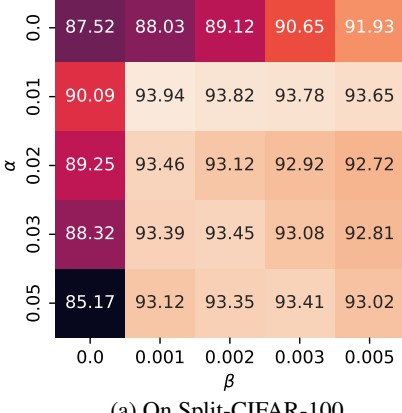
(a) On Split-CIFAR-100

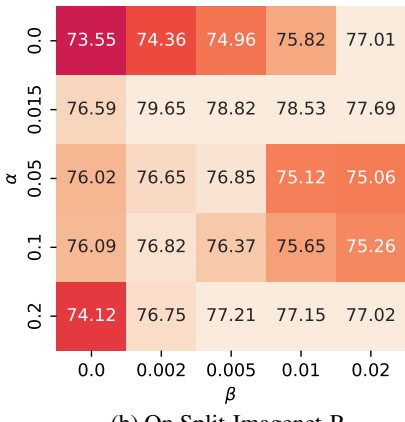
(b) On Split-Imagenet-R

Figure 17: Model performance when varying the value of $\alpha, \beta$.

### E.7 DISCUSSION

**Limitation.** Despite our novel approach, the quality of the hierarchical taxonomy can depend on the quality of expert knowledge, thus affecting the model performance. For example, if similar image classes are not assigned to the same leaf group in this label-based taxonomy, the constraint we put on each such group may not perform as expected. Therefore, in our experiment, we utilize the latest power full LLMs to support this strategy, thereby mitigating the possibility of inappropriate organization which can harm performance. See Appendix E.1.

**Potential impacts.** In this paper, we aim for Continual Learning for Image Classification tasks. However, this method may promise to effectively support more complex tasks in Computer Vision, such as Object Detection, Action Recognition, etc., in general. Furthermore, organizing concepts into groups may also hold promise for knowledge transfer strategies for CL settings, helping the models flexibly reinforce relevant knowledge according to each specific domain, which we will reserve for future work.

