# OpenReview forum: "Reflecting and Linking knowledge: Dynamic Label Structures for Prompt-based Continual Learning"
_ICLR.cc/2026/Conference — ICLR 2026 Conference Withdrawn Submission_

### Official Review · Reviewer_6dwa · 2025-10-24

**Soundness:** 1
**Presentation:** 1
**Contribution:** 1
**Rating:** 2
**Confidence:** 5

**Summary:**

This paper studies Continuous Learning (CL), with a particular focus on prompt-based methods. Its core idea is to imitate the way humans learn and solve the "catastrophic forgetting" problem in deep learning by organizing and correlating old and new knowledge.
It identifies the most confusing category combinations by dynamically building old and new category labels into a tree-like hierarchy (taxonomy). Then, the model will pay special attention to distinguishing these similar categories during training, thereby effectively reducing forgetting and improving learning results.

**Strengths:**

The research topic of class incremental learning is reasonable.

**Weaknesses:**

1. Limited technical contribution. Using hierarchical taxonomy has been proposed in [1]. The idea of supervised contrastive losses is similar to [3]. The proposed method seems like a combination of existing methods.
2. Unrealistic and Impractical: This approach is entirely impractical. It assumes for any given CL stream, one has access to a powerful LLM and a carefully engineered prompt to generate a useful taxonomy on the fly. This introduces significant overhead and a major dependency that is not part of the core learning model. This is not a "limitation," as the authors weakly claim in the conclusion; it is a fundamental design flaw that makes the entire enterprise scientifically unsound in the context of continual learning. The fact that this crucial detail is buried in the appendix and a disclosure statement, while the main paper vaguely refers to "experts," is a serious presentation issue. I don't think it's necessary to introduce LLM in CIL.
3. The experiments are insufficient; the author should compare with more SOTA CIL methods, including RanPAC, TUNA, etc. The compared methods are not limited to prompt-based methods.

4. The writing is poor and unprofessional. The paper is replete with hand-wavy appeals to "Cognitive Science," "human learning," and "reflection." These are used as post-hoc justifications rather than foundational principles that guide the model design. Phrases like "uncover fresh perspectives," "dive deeper into the hidden connections," and "from these revelations" (Abstract) are unscientific and belong in a blog post, not a top-tier conference paper.
The connection to "reflection" (Schön, 1983) is tenuous at best. The model does not "reflect"; it is simply handed a pre-computed structure by an external agent. This is a gross mischaracterization of the proposed algorithm.
5. T-SNE Visualizations are Anecdotal: The t-SNE plots (Fig. 3, 14-16) are classic examples of "pretty pictures" that provide little scientific insight. Of course, applying a stronger contrastive loss to specific groups will make those groups more separated in a 2D projection. This does not prove the method is better or that this separation is the optimal thing to do for downstream accuracy.

6. This paper lacks citations, comparisons, and discussions of some necessary articles, including [1]-[5].

[1] ChatGPT-Powered Hierarchical Comparisons for Image Classification

[2] Enhancing Visual Continual Learning with Language-Guided Supervision

[3] Prototype Augmentation and Self-Supervision for Incremental Learning

[4] Integrating Task-Specific and Universal Adapters for Pre-Trained Model-based Class-Incremental Learning

[5] Ranpac: Random projections and pre-trained models for continual learning

**Questions:**

NA

---

### Official Review · Reviewer_Ngsz · 2025-10-29

**Soundness:** 3
**Presentation:** 3
**Contribution:** 3
**Rating:** 4
**Confidence:** 2

**Summary:**

This paper addresses catastrophic forgetting in prompt-based continual learning, arguing that forgetting is exacerbated by the uncontrolled overlap of class representations in the latent space as new, semantically similar classes are introduced. To address this, the authors propose a method that organizes class labels into a dynamic hierarchical taxonomy. This taxonomy is then used to define a new regularization loss that applies a stronger "inner-group" contrastive force to classes within the same leaf, thereby focusing on separating classes that are easily confused. The method also introduces a refinement using optimal transport to measure the distance between class priors in the pretrained model, which then weights the contrastive loss to push initially closer class pairs further apart. The results show strong performance, but the core idea relies on external knowledge to build the hierarchy.

**Strengths:**

-   The paper identifies a very clear and plausible mechanism for forgetting: the increasing density and overlap of representations for semantically similar classes as tasks accumulate.
-   The core idea of using a label hierarchy to apply a more focused, "inner-group" contrastive loss is intuitive and well-motivated. It makes sense to spend more effort separating "oak tree" from "pine tree" than from "mouse".
-   The experiments are thorough and the presentation is easy to follow. The method is benchmarked on four datasets and consistently outperforms a strong set of recent, prompt-based baselines.

**Weaknesses:**

-   The method's primary component, the "dynamic label-based hierarchical taxonomy", is not learned from the data but provided by an external "expert". The appendix reveals this expert is a large language model like GPT-4. This is a very significant, external dependency that is not part of the learning model itself and feels like it sidesteps the difficulty of discovering these relationships.
-   This reliance on an external, powerful oracle (the LLM) to provide the class structure at each step changes the problem setting from a standard self-contained continual learning scenario. The model's performance is now heavily dependent on the quality of this external knowledge.
-   The paper claims the taxonomy is built "incrementally", but the prompts shown in Appendix C suggest the tree is re-generated from the *full list* of labels seen so far at each task, which is not an incremental process. This seems computationally inefficient and relies on having all past labels available.
-   The optimal transport refinement adds significant complexity, requiring GMM fitting for all classes and $W_2$ distance calculations. However, the ablation study (Figure 5b, Table 5) shows it provides a relatively small performance boost over just using the taxonomy-guided loss alone. This brings into question the practical value of this component.

**Questions:**

-   Could the authors provide more clarification on the role of the LLM? How critical is the "quality" of the taxonomy? For instance, what would the performance be if a simpler, non-LLM-based hierarchy (like one derived from WordNet, or simple k-means clustering on features) were used instead? This would help separate the benefit of the loss mechanism from the benefit of having a perfect, external knowledge source.
-   The paper describes the taxonomy construction as "incremental". Could you clarify if this is the case? The appendix example seems to show a prompt that re-generates the entire tree using the full list of seen labels, which would not be a truly incremental or scalable process.
-   The optimal transport-based weighting (Eq. 4) appears to offer a marginal gain based on the ablation studies (e.g., in Table 5, (C) vs (D)). Given the added complexity, do the authors see this as an essential component, or is the primary contribution from the main taxonomy-guided contrastive loss?
-   In Section 4.2, GMMs are used to sample features for old classes. How sensitive is the method to the number of components K chosen for the GMMs?

---

### Official Review · Reviewer_U8V3 · 2025-11-02

**Soundness:** 3
**Presentation:** 4
**Contribution:** 3
**Rating:** 6
**Confidence:** 4

**Summary:**

This paper targets rehearsal-free, prompt-based class-incremental learning (CIL) and argues that, beyond parameter interference and prompt–selection mismatch, a major yet under-addressed source of forgetting is representation-level entanglement between semantically close classes that arrive in different tasks. To mitigate this, the authors propose RefCL, which (i) incrementally builds a dynamic, label-based hierarchical taxonomy to identify “high-confusion” leaf groups when new classes arrive, and (ii) trains with a group-aware supervised contrastive objective that puts extra pressure on separating classes inside those leaf groups. On top of that, they further reweight the group contrastive loss using OT/Wasserstein-2 distances computed from the frozen pretrained backbone, so that pairs of classes that the backbone deems closer get stronger push-apart signals. Experiments on typical prompt-based CIL benchmarks (Split CIFAR-100, ImageNet-R, CUB-200, 5-Datasets) against strong prompt baselines such as L2P, DualPrompt, HiDe-Prompt, etc., show consistent gains in Final Average Accuracy and reduced forgetting.

**Strengths:**

1. Pinpoints a concrete, under-explored forgetting source. Most prompt-based CL papers frame forgetting as “shared prompt/backbone overwritten” or “prompt retrieval mismatch” (see L2P, DualPrompt, HiDe-Prompt). This paper makes the more specific claim that cross-task, semantically-close classes collapse in representation space and that this collapse materially contributes to forgetting. That is a sharper diagnosis than the usual “stability–plasticity” catch-all.
2. Hierarchy-as-a-lens for CIL is novel in this sub-line. Compared with existing prompt CL variants that improve retrieval, prompt discretization, or language guidance (LGCL), this work’s idea of dynamically expanding a label taxonomy as tasks arrive and then using it to decide which classes deserve stronger separation is a fairly original control knob for rehearsal-free CIL.
3. OT-based difficulty reweighting is a nice, model-aware twist. Instead of hand-picking “hard” class pairs, they use the initial pretrained backbone itself to estimate inter-class similarity via W2/OT, and then translate this into contrastive weights. That makes the method self-referential (“reflecting” on the model’s own geometry) and is more principled than uniform group contrast.
4. Empirical story is consistent with the claim. On exactly the benchmarks where cross-class similarity is a real problem (CUB-200, ImageNet-R), they report gains over the same prompt baselines other 2024–2025 papers also compare to (L2P, DualPrompt, CODA-Prompt, HiDe-Prompt, VQ-Prompt), so the improvement does not look like an artifact of a weak baseline.

**Weaknesses:**

1. Taxonomy construction is the softest part of the story. The whole method stands on “we can incrementally form sensible label-based groups,” but the paper does not make this part fully algorithmic or provably robust. If the grouping is manual / heuristic / LLM-assisted, then (i) reproducibility across labs is questionable, (ii) performance may silently depend on domain knowledge of the annotator, and (iii) the approach becomes less convincing than works that learn or rearrange class space end-to-end (e.g. RCS-Prompt) for the same goal of reducing confusion.
2. OT on frozen features can be a biased teacher. Their weighting assumes that the original pretrained backbone’s geometry correctly reflects future, task-conditioned difficulties. But prompt-based CL precisely deforms that geometry task by task; if the initial backbone under-separates some fine-grained classes (very plausible on CUB-200), the OT weights will amplify this early bias instead of correcting it. There is no analysis of error propagation from “wrong OT distances → wrong weights → over-regularization on the wrong pairs.”
3. Closest/rival ideas are not confronted head-on. In 2023–2025 there are several papers that also inject structure / guidance into prompt CL — e.g., LGCL (language-level class structure), RCS-Prompt (rearranging class space), and more recent prompt-selection correctness papers. The submission cites mainstream prompt baselines, but it’s not obvious from the current text whether RefCL truly dominates these “structure-aware” contemporaries when all are run on the same backbone and task order. That makes the “we are the first to tackle class entanglement explicitly” message weaker.
4. Computational overhead and scaling not fully justified. The method needs (a) maintaining/expanding a taxonomy every task and (b) computing pairwise OT/W2 distances between class distributions at each step. For small CIL benchmarks this is fine, but for realistic 200–1K class streams this becomes (O(C^2)) in the number of active classes unless you prune. The paper does not explain how to cap this or to approximate OT while keeping the claimed benefits.

**Questions:**

1. How brittle is RefCL to *wrong* or *coarse* hierarchies? Suppose I deliberately merge two visually dissimilar classes into one leaf group, or I fail to merge two actually-confusable ones. Does the method degrade gracefully (just lower gains) or can it actually hurt past-task accuracy because the contrastive head is now pushing apart the wrong instances? I would like to see a synthetic “taxonomy noise” experiment (0%, 25%, 50% wrong merges). Right now the method implicitly assumes “we can build good groups.”
2. What is the exact asymptotic/per-task cost of OT-based reweighting, and can it be amortized? In the presented form, every new task may require recomputing pairwise W2 between the new classes and all past classes. On CIFAR-100 this is fine, but on larger or longer streams this is expensive; also, how often do they recompute GMMs, and are those GMMs fit on actual images or on stored feature summaries? A complexity table would make the method much more convincing for real-world incremental setups.

---

### Official Review · Reviewer_6G35 · 2025-11-02

**Soundness:** 3
**Presentation:** 4
**Contribution:** 2
**Rating:** 4
**Confidence:** 4

**Summary:**

This paper proposes new regularizations to be included along with cross-entropy loss to mitigate catastrophic forgetting in prompt-based continual learning for image classification tasks. Their approach titled RefCL is cognitively inspired and is motivated by how humans organize knowledge hierarchically and reflect on connections between old and new information. To that end, they propose to introduce a dynamically evolving label-based hierarchical taxonomy, built incrementally as new tasks appear. The hierarchy is constructed by grouping semantically and visually similar classes under shared leaf nodes. Since similar classes  in the same group could create higher confusion, they propose to impose a stronger separation  between similar classes  through a supervised contrastive loss  while still maintaining discrimination across all classes (the “outer-group” constraint).
Next, to address the absence of past data in class-incremental learning, the authors learn a generative model based on GMM  on the latent representation of each class. They then sample from this model to get synthetic anchors for old classes when computing the supervised contrastive losses. To further differentiate similar classes in a group, they also introduce an Optimal Transport (OT)-based weighting scheme that leverages the pretrained model’s latent geometry: pairwise Wasserstein-2 distances between class distributions define weights that emphasize confusable class pairs. This ensures that overlapping representations in the pretrained feature space receive higher penalty.  They conducted extensive experiments on different benchmarks and a show consistent gains of 1–2% in final average accuracy and also strong improvement in reteniton/ less forgetting of previous classes. The t-SNE visualizations show clearer class separations within semantically close groups, and ablations confirm the additive benefits of the hierarchical and OT-based components. Authors also demonstrate that various LLMs (GPT-4o, Llama-3, Claude, Gemini) produce similar taxonomy structures and comparable results, showing the robustness of LLM-assisted grouping in lieu of expert grouping.

**Strengths:**

The paper presents a simple but well-motivated regularization approach for continual learning using hierarchical relationships between labels. The use of semantic groupings to highlight confusable classes is intuitive and directly aligned with how representation overlap happens in pretrained ViTs. The integration of an OT-based weighting step on top of these groupings is novel and makes good use of  latent geometry without retraining the backbone. The method remains rehearsal-free and only adds lightweight regularization terms on top of existing prompt-based pipelines. The experiments are comprehensive, spanning multiple datasets and strong baselines, and the visualizations help interpret the improvement. The ablation studies are carefully done and clearly support the design choices.

**Weaknesses:**

Before proceeding to enumerate my concerns regarding the methodology, the reviewer would like to express their issue regarding the scope of the problem. The current formulation is tied entirely to image classification tasks, which strongly limits the broader applicability of the idea. Given that continual learning challenges appear across domains including LLMs (pre-, mid-, and post-training) as well as multimodal training setups where task boundaries or taxonomies cannot be easily constructed, it is unclear how this idea would translate, especially given that the motivation is cognitively inspired. Even for vision, tasks such as segmentation or object detection do not have simple label hierarchies, making the current method less general. Connected to this, the method is also limited to supervised cross-entropy–based classification, while the field has made significant progress toward zero-shot classification as done in CLIP, even for downstream classification tasks where the CLIP model needs to be fine-tuned (see https://arxiv.org/pdf/2212.00638). One could argue that since the hierarchy is textual, it would have been more appropriate to use a multimodal zero-shot classification model and study or improve its behavior in continual learning settings. In fact, the authors do not discuss such methods in any detail. Lastly, from the scope perspective, the authors consider only one setting of CL (class-incremental learning) and one class of methods (prompt-based). The proposed principles could even be at odds with domain-incremental learning since they introduce another level of semantic grouping that may not align with domain boundaries.
—
It is also not immediately clear how robust the method is to imperfect or inconsistent groupings (the reviewer is aware of Appendix E.1). In realistic applications that go beyond well-defined class names like CIFAR-100, experts or LLMs may and will generate noisy or incorrect hierarchies. The reviewer acknowledges that the authors list this as a limitation, but it would be prudent to study robustness explicitly. To that end, a small controlled experiment could be added where the taxonomy is intentionally corrupted (e.g., misplacing “willow tree” under “four-legged animals”) to test sensitivity to grouping errors.   In a similar vein, what happens if no two classes can be semantically grouped? Would RefCL’s regularization have any effect, or would it converge to a baseline? In the limiting case where every class is its own leaf node, since the regularization targets only the third cause of forgetting listed in Section 3.2 (representation overlap), the method should not yield further benefit. The reviewer is not making a reductive argument but rather emphasizing that additional regularization strategies may still be needed to make RefCL effective beyond its assumed setting. Clarifying these aspects in the rebuttal would be helpful.

Next, the OT-based weighting relies on Wasserstein-2 distances, yet Figure 4 suggests these distances are unnormalized. Consequently, the statement in Line 296 “the distance between mouse and porcupine is significantly larger than between mouse and hamster” appears too strong. It is unclear how “significance” is defined here, as distances like 26 and 32 may not be meaningfully different if the range is unbounded. This might also explain the sensitivity to the parameter δ in Figure 6.

The use of GMMs as class-level generative models also needs further justification. Is the choice purely driven by computational efficiency? GMMs are parametric density estimators, and while they allow direct sampling, the resulting samples are not guaranteed to represent high-density regions of the true latent distribution. Without likelihood-based filtering or rejection, some samples may lie in low-probability regions, especially when the mixture fit is coarse. Would RefCL benefit from more accurate sampling or improved generative modeling? Furthermore, to construct an oracle version of RefCL, what would happen if one used richer generative models such as GANs, as explored in earlier continual learning literature?

Finally, the motivation of this paper is thematically related to works such as SACK by Kundargi et al. (https://openreview.net/notes/edits/attachment?id=4oAi8pvXjR&name=pdf) and the recent work on Language-Guided Concept Bottleneck Models for Interpretable Continual Learning (https://openaccess.thecvf.com/content/CVPR2025/papers/Yu_Language_Guided_Concept_Bottleneck_Models_for_Interpretable_Continual_Learning_CVPR_2025_paper.pdf), which study class similarity and representation reorganization through the interpretability lens. It would be useful to discuss the relationship between these lines of work and RefCL in terms of how interpretability-driven grouping compares to the proposed taxonomy-based structuring.

**Questions:**

Please refer to the weakness section.

---

### Note · Authors · 2025-11-14

I have read and agree with the venue's withdrawal policy on behalf of myself and my co-authors.